# A Learning-Based Framework for Fair and Scalable Solution Generation in Kidney Exchange Problems

**William St-Arnaud**                                                  *william.st-arnaud@umontreal.ca*
*Université de Montréal*
*Mila*

**Margarida Carvalho**                                                 *carvalho@iro.umontreal.ca*
*Université de Montréal*
*Mila*

**Golnoosh Farnadi**                                                   *farnadig@mila.quebec*
*McGill University*
*Mila*

**Reviewed on OpenReview:** *https://openreview.net/forum?id=IizmQoF86Y*

## Abstract

Reinforcement learning and Generative Flow Networks, known as GFlowNets, present an exciting possibility for neural networks to model distributions across various data structures. In this paper, we broaden their applicability to data structures consisting of optimal solutions for a combinatorial problem. Concretely, we propose using *Q*-learning and various policy gradient methods, as well as GFlowNets to learn the distribution of optimal solutions for kidney exchange problems (KEPs). This could provide a useful tool for decision-making authorities, policymakers and clinicians, as it offers them multiple optimal or near-optimal solutions, and provides a complementary landscape to their traditional integer programming-based toolbox for promoting fairness and societal benefits. Our reinforcement learning-based framework trained on KEP instances provides an effective addition to computationally expensive exact approaches, notably mixed-integer programming. Our experiments thoroughly evaluate the quality of the solution sets sampled from the trained neural networks in terms of optimality, their scalability when dealing with real-sized KEP instances, and their capability to generate a diverse pool of solutions. We also cover the use of their efficient solution generation capabilities to improve fairness and simulate the evolution of the KEP pool in a dynamic setting. Our contribution is thus: 1) methodological, as it introduces a novel setting for reinforcement learning in addition to GFlowNets, 2) implementational, as it delves beyond the theory and details how to use conditional information, and 3) of practical significance, as it considers a specific combinatorial problem in the healthcare domain.

## 1 Introduction

Resource allocation problems are ubiquitous in our everyday lives as they find various applications in sectors such as business, engineering, robotics, and healthcare (Cormen et al., 2022; Bin-Obaid & Trafalis, 2020; Munguía-López & Ponce-Ortega, 2021). Many of these problems take the form of combinatorial optimization problems due to the indivisibility of resources, which makes them hard to solve. One large class of such problems consists of matching problems. In the healthcare sector, kidney exchange problems (KEPs) are well-known matching problems involving patients needing a kidney transplant and donors. The goal is to find an exchange plan (defined in section 2), which determines the patients receiving a transplant. The objective driving the search for an exchange plan can vary but often involves the maximization of some

objective function taking into account the number of matched patients (i.e., receiving a transplant) among other criteria.

Undoubtedly, KEPs correspond to an important subset of resource allocation problems in combinatorial optimization. In this context, we identify three key challenges. First, the exchange plans that are selected can sometimes fail for logistical reasons and due to donor or patient drop-off, e.g. a patient's health status renders them unable to perform surgery. Second, KEPs often have multiple optimal exchange plans, and relying on exact solvers may not ensure all patients have a fair chance of receiving a transplant. Third, simulating KEPs in the long term is computationally expensive, notably, because each round of a KEP corresponds to a hard combinatorial problem. Because current KEPs use ad-hoc or empirically-based objectives to guide their matching policy, such exchange systems have yet to fully demonstrate the long-term impact of derived policies on the patient pool. For example, it has been observed in certain KEPs that there is an accumulation of certain subgroups of patients in the pool, based on bloodtype (Canadian Institute for Health Information, 2016).

Based on the first two aforementioned challenges, efficiently computing multiple exchange plans would empower decision-makers with the ability *(i)* to dispose of alternative exchange plans in situations of unexpected infeasibility, and *(ii)* to draw exchange plans at random according to a fixed policy and guarantee probabilities to patients of receiving a transplant, akin to live lotteries for citizens' assemblies (Flanigan et al., 2021). With respect to the third challenge posed above, the efficient simulation of matching policies would *(iii)* afford decision-makers the ability to discover long-term policy impacts and mitigate potential negative effects. Because of the delicate nature of allocating kidneys to patients, the ultimate decision-making power lies in the hands of policy-makers and clinicians. They stand to gain greater advantages from a diverse array of exchange plans rather than relying on a singular one.

Because KEPs are usually formulated as mixed-integer programs, distributions of solutions can be generated through column generation (St-Arnaud et al., 2023). For large kidney exchange pools, the mixed-integer formulation can become impractical to describe, and the time to compute one solution, not to mention a distribution of solutions, can be prohibitively large. To complement these approaches, reinforcement learning (RL) methods can learn policies that build exchange plans sequentially. The learned models can then be used to generate solutions. Because of the need to scale to large kidney exchange pools and to model long-term behaviour of policies dictating the selection of exchange plans, it is important to better understand the pros and cons of using learning-based methods to derive KEP policies. In this work, we explore the possibility of efficiently generating multiple optimal or near-optimal solutions (exchange plans) by learning policies to generate exchange plans. This fits neatly into the framework of reinforcement learning through policy optimization methods. These include policy gradient, promximal policy optimization (PPO), and $Q$-learning. By deriving the policy directly (and indirectly through $Q$ values in $Q$-learning), the possibility of sampling multiple high-reward solutions arises. More recently, *generative flow networks* (GFlowNets), forming a new class of algorithms to sample solutions proportional to some reward (much like PPO), have shown promising results (Bengio et al., 2023; 2021). Our first contribution is the application of these learning-based methods to KEPs with the goal of using an extended set of solutions to form probability distributions over exchange plans. Second, to fit into the reinforcement learning and GFlowNet framework, we formalize the states, actions, trajectories, episodes and reward function that define the sampling distribution to be learned. Moreover, we condition on the input KEP graph as it allows us to model a rich family of distributions from which we can sample. Third, we experimentally demonstrate[1] the ability of our learning procedure to efficiently output good-quality (near-optimal) distributions by comparing against baselines, including the selection of an optimal solution. In addition, we analyze the capability of the learning phase to generalize from smaller KEP instances (i.e. training set) to larger instances (i.e. test set) when computing one or multiple exchange plans.

The promising experimental results validating the use of RL and GFlowNets enable us to advance to our fourth contribution: showcasing the value of having an efficient sampler downstream of the training phase. To this end, *(1)* we provide an effective sampling mechanism through which decision-makers can report individual matching probabilities to patients in the KEP pool, and *(2)* we describe the use of our approach

---

[1]Our code will be made available upon publication.

within the simulation of the evolution of the KEP pool over time. In *(1)*, by using exchange plans generated through policy learning, we demonstrate how to improve *individual fairness* (IF) (i.e. more equal probability of receiving a transplant over patients) measures by computing a diverse set of (near-)optimal solutions. We compare against an optimal (but not scalable) exact mixed-integer programming approach. In *(2)*, we use our efficient samplers and we show that their use in a dynamic simulation of the kidney exchange system allows to approximate the expected number of transplants over multiple matching rounds. These two experiments reflect how the work developed in this article can empower decision-makers to assess fairness thanks to the availability of multiple optimal solutions and to better understand the long-term impact of their choice of matching policy on patient welfare.

In section 2 of this article, we introduce key definitions and concepts relating to KEPs. We follow with a literature review of solution approaches and fairness for KEPs, machine learning in combinatorial optimization and reinforcement learning approaches for sampling solutions. In sections 4 and 5, we detail all the necessary concepts for learning-based methods applied to KEPs. Section 6 contains the experimental setup for our research questions. In sections 7 and 8, we introduce research questions and we detail the results by providing a thorough analysis. Finally, section 9 contains our concluding remarks and we highlight future research directions.

## 2 Kidney exchange

With the purpose of addressing the scarcity of kidneys on deceased donor waiting lists, Rapaport (1986) put forward the notion of living donor exchanges. This led to what is now known as kidney exchange (Roth et al., 2004), a barter market that has been implemented in several countries, namely, in Europe (Biró et al., 2021), Canada (Malik & Cole, 2014) and South Korea (Park et al., 2004).

In a kidney exchange system, incompatible patient-donor pairs are registered, forming a KEP pool. These pairs are then matched with other incompatible patient-donor pairs to exchange donors, resulting in compatible transplants to be performed. Matching between pairs can be done either pairwise or involve multiple pairs as part of a *cycle*. Additionally, non-directed donors can also register in the system, without being attached to a specific patient, and allow exchanges to initiate on them, thereby leading to *chains*. Indeed, a KEP can be naturally modeled as a graph $G = (V, A)$; see Figure 1. In Figure 1, the set $P$ of patient-donor pairs is represented by vertices and an arc $(u, v)$ between vertices $u$ and $v$ corresponds to compatibility between the donor of pair $u$ and the patient of pair $v$. The set $N$ in Figure 1 consists of non-directed donors, here the grey-coloured vertices. The vertex set of the graph is equal to the disjoin union of $P$ and $N$, i.e., $V = P \cup N$ with $P \cap N = \emptyset$. Notice that there are no incoming arcs to vertex $v_8$. In this example, we can observe cycles involving vertices from $P$, e.g. $(v_2, v_7, v_4)$, and chains starting in vertices from $N$ and extended with vertices from $P$, e.g. $(v_8, v_3, v_2, v_5)$ where $(v_8, v_3, v_2)$ is a subchain that is also a valid chain. We use the general term *exchange* to refer to a cycle or a chain. Consequently, solving a KEP entails determining an exchange plan, i.e., a set of disjoint exchanges (matchings) as only one kidney can be transplanted from a donor. In addition, for logistic reasons, the list of allowed exchange plans is limited to those including cycles and chains of below a predetermined length. This constraint is behind the complexity of solving KEPs, as they would otherwise be tantamount to assignment problems. In Figure 1, an exchange plan, where the maximum predetermined exchange length is three, takes the form of a subgraph of the original KEP graph (see subgraph on the right in the figure). The task of finding an exchange plan is usually approached through the use of solvers optimizing an objective, such as the maximization of the total number of transplants, the so-called *utilitarian objective*. Thus, in kidney exchange systems, at each predefined interval (e.g., every four months), a KEP is solved to find an exchange plan among the currently registered patients and donors that optimizes the objective function. This over-time setting is referred as a dynamic KEP in the remainder of the paper.

## 3 Literature review

**Solving KEPs.** In its most basic form, KEPs entail optimizing the utilitarian objective, which is already NP-hard (Abraham et al., 2007). The simplest mixed-integer program (MIP) to describe a KEP is the

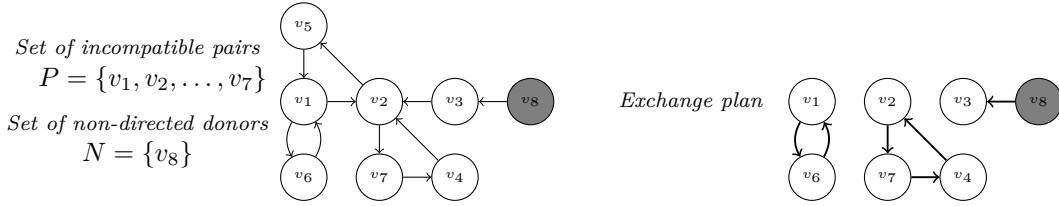

Figure 1: On the left, an example of a KEP graph. On the right, an example of a feasible exchange plan for a predetermined maximum exchange length of three.

*cycle formulation* (Abraham et al., 2007; Roth et al., 2007). Other MIPs of a more compact size have been proposed, for example, by Constantino et al. (2013) and Dickerson et al. (2016). MIPs are widely used since they can be fed directly to off-the-shelf solvers. While the usual goal in KEPs is to identify a single optimal exchange plan according to some pre-agreed objective, other works have proposed the determination of a probability distribution over exchange plans, accounting for fairness during selection. For instance, Farnadi et al. (2021) enumerate all optimal solutions for the utilitarian objective in order to generate a probability distribution across them from which to sample exchange plans. This approach is better known as *individual fairness*, and it only scales for small instances with at most 50 pairs. Next, we discuss fairness for KEPs in more detail.

**Fairness for KEPs.** It was observed by Dickerson et al. (2014) that solely maximizing the number of transplants in KEPs can result in unfair treatment of specific patient groups that accumulate in the KEP pool over time (c.f. RQ4 in section 8). For this reason, there has been substantial research on group fairness approaches for selecting an exchange plan (e.g., Dickerson et al. (2014); Dickerson & Sandholm (2014); Gao (2019)). Recent work has introduced a distinct fairness concept—individual fairness—which focuses on ensuring fairness at the level of individual patients rather than patient groups. This idea involves offering decision-makers multiple feasible solutions, moving away from the traditional focus on a single solution returned by solvers (Farnadi et al., 2021; St-Arnaud et al., 2023; Demeulemeester et al., 2025). St-Arnaud et al. (2023) obtain probability distributions through the enumeration of solutions in the support of multi-objective KEPs composed of utilitarian and fairness components. The drawback of their methodology is that the choice of a specific fairness objective is predetermined before solving the optimization problem. This restricts the solution landscape to those that align with the chosen objective, potentially excluding other fair or efficient solutions that policy-makers might wish to explore.

**Machine learning and combinatorial optimization.** Across the literature on combinatorial optimization problems (which includes MIPs), a commonly found solution approach relevant to our work involves the use of machine learning models and architectures to return high quality solutions. This method belongs to a subtype of *end-to-end constrained optimization* known in the literature as *predicting solutions* (Kotary et al., 2021). Specifically, Lagrange multipliers have been used in learning objectives to enforce feasibility of constrained problems (Hopfeild & Tank, 1985), while Detassis et al. (2021) introduced an iterative algorithm that uses a combinatorial optimization solver to adjust solutions returned by a neural network in order to maintain feasibility. Vinyals et al. (2015) introduced pointer networks to solve the travelling-salesman (TSP) and convex-hull problems, but their method applies to instances where all permutations of a solution are feasible; this is the case for KEP cycles, but not for chains. RL methods such as the actor-critic framework have been used to solve combinatorial optimization problems by relaxing the optimality criterion of generated solutions (Bello et al., 2016). More recent advances in Velickovic et al. (2018) use graph attention networks with REINFORCE (Williams, 1992) to improve upon previous results on tasks such as TSPs.

A major driving factor behind this line of research comes from the inherent difficulty in solving some large combinatorial optimization problems exactly; in some well-known cases, an approximate solution that is feasible is satisfying (e.g. chess, TSP), thus further justifying the use of learning-based methods. Moreover, when combinatorial optimization problems involve uncertainty, it is typically intractable to solve a problem exactly and it is then natural to use learning-based methods. Other learning-based methods are available, such as *ML-augmented constrained optimization* and *predict-and-optimize* (Kotary et al., 2021). The former

guide a combinatorial solver during the solution phase, while the latter learns to predict a proxy model (objective and/or constraints) that is fed to a combinatorial optimization solver, which returns a solution that is ideally close to a target solution (Ferber et al., 2020; Aubin-Frankowski et al., 2024). Much like MIP, if the combinatorial formulation of the problem or the time to recover recover a solution is prohibitively large, these methods can fail. While predicting solutions has shown promise in certain tasks such as portfolio optimization and mixed-integer linear programming (Wilder et al., 2019; Ferber et al., 2020), we observe that these learning-based methods, much like MIP, are most commonly used for selecting one solution, in contrast to our goal of producing a full distribution of solutions. The main difficulty with the aforementioned methods emerges out of the training procedure, but once a model is learned, we can often get an efficient sampler that allows for fast solution generation. Based on this idea, we describe in the paragraph below how we seek to take advantage of this potentially highly efficient solution generation to address fairness concerns raised earlier in relation to MIPs.

**Solution sampling in RL.** In this article, we seek to investigate learning methods that can sample exchange plans efficiently. We are motivated by the need to find distributions of exchange plans exhibiting certain properties, as it is the general case for fair assignment problems involving the allocation of critical resources. Our work is also driven by the fact that such mechanisms can allow the efficient simulation of multiple matching rounds (dynamic KEP). The key aspect that differentiates our line of inquiry from prior research on KEPs is the learning aspect involved, where one is looking to learn a distribution of optimal (or near-optimal) solutions, according to a given objective. For this reason, we focus on policy gradient and flow-based methods (Sutton et al., 1999; Bengio et al., 2023), and extend their application to the setting of kidney exchanges. These approaches are well suited for application on discrete structures such as graphs as it is possible to enforce constraints within the set of possible actions. In addition, GFlowNets offer in some cases an improvement over policy-gradient methods and Monte Carlo Markov chain (MCMC) sampling, because of their ability to model the reward directly and learn modes of the distribution. Algorithms derived from both policy gradient and generative flow networks have previously been successfully used in drug discovery (Bengio et al., 2021; Angermueller et al., 2019). The setting in drug discovery is similar to ours in the sense that it seeks to output samples that bear a particular graph structure (drug-like molecule). Yet, it is also different, since the output in our problem is conditional on an initial KEP graph (see section 5), i.e. the action and state spaces are countably infinite (see section 6).

## 4 Learning-based methods

We proceed to define the usual concepts of RL such as state, action and reward, in the setting of static KEPs. We will later see in section 5 the process by which we learn static KEP policies and how these policies are deployed over *episodes* consisting of multiple matching rounds with uncertainty over the arrival and departure of patient-donor pairs.

**Preliminaries.** As argued in the previous sections, learning-based methods (e.g. policy gradient and GFlowNets) seem to offer an adequate approach for learning probability distributions aimed to address fair assignment problems, namely, KEPs. In our application of learning methods to KEPs, we propose to sequentially build exchange plans by adding cycles and chains until a maximal exchange plan is reached. Thus, given a set of disjoint cycles and chains (hence a KEP subgraph, or exchange plan), the policy to be learned will assign probabilities to the remaining feasible (i.e., disjoint) cycles and chains. A cycle or chain is selected by drawing from the constructed distribution. Note that $Q$-learning does not directly offer a probability distribution, but we can recover one by making use of a $\epsilon$-greedy selection mechanism. This approach involves either selecting uniformly at random between a feasible cycle or chain, or choosing the one with the maximum value (i.e. maximum $Q$-value).

In the process of building an exchange plan, two scenarios are possible: the current exchange plan is complete (i.e., maximal) or not yet. In the case where it is not yet maximal, we augment it with additional cycles and chains. In the case where it is maximal, the only available option is to select the exchange plan and proceed to the next matching round. Because of these two possible scenarios, we will define two types of states, **internal** and **external** states.

**Definition 4.1** (State). General states, denoted with the variable $s$, will consist of a tuple $s = (G, H)$, where $G$ is a KEP graph and $H$ is an exchange plan, i.e. a KEP subgraph of $G$ consisting of disjoint cycles and chains. The set of parent and children of state $s$ will be denoted using $par(s)$ and $child(s)$, respectively. An **internal state** $s = (G, H)$ is a state such that for all $s' \in child(s)$, there exists $H'$ such that $s' = (G, H')$ and $H$ is a subgraph of $H'$. An **external state** $s = (G, H)$ is a state such that for all $s' = (G', H') \in child(s)$, we have $H' = \emptyset$. We denote the set of external states as $\mathcal{X}$.

In the case of an internal state, an (internal) action (see Definition 4.2 below) will correspond to cycles and chains that can be included to augment the exchange plan in the internal state (i.e., its second component). Specifically, (internal) actions are arcs between an internal state $s$ and a state $s'$, such that $s$'s exchange plan is a subgraph of $s'$'s exchange plan. When convenient, we will denote such arcs as $a = (s, s')$. In the case of an external state, an (external) action corresponds to the selection of the exchange plan as a matching. Specifically, (external) actions are arcs between two states $s$ and $s'$, such that $s'$'s exchange plan is $\emptyset$. We can also think of internal states as part of the matching-building process in a specific matching round, while external states correspond to the end of a matching round.

**Definition 4.2** (Action). A general action is an arc $a = (s, s')$ between two states. It is an **internal action** if $s = (G, H)$ and $s' = (G, H')$ implies that $H$ is a subgraph of $H'$. It is an **external action** if $s' = (G', \emptyset)$ for some KEP graph $G'$.

**Definition 4.3** (Transition function). The transition function $T$ is implicitly defined as $(s, a) \mapsto s'$ with probability 1 for $a = (s, s')$ when $s$ is an internal state. When $s = (G, H)$ is an external state, we have a random variable $M_G$ such that $s' = (G', \emptyset) \sim M_G$, where $G'$ is obtained through the graph $G$ after the arrival and departure of patient-donor pairs and non-directed donors in the pool using a Markovian process $\mathcal{M}$.

The transition function corresponds to adding cycles and chains for internal states and transitioning to a new KEP graph after a matching round for external states.

**Definition 4.4** (Reward). At each state, the reward for $s = (G, H)$ and $a = ((G, H), (G', H'))$ is given as $R(s, a) = e^{|P(H')|}$ for a maximal matching $H'$ (see Definition 4.5) and 0 otherwise, where $P(H')$ is the vertex set of $H'$ and it corresponds to the set of matched patients.

Using this representation of states and actions, it is possible to form a directed acyclic graph (DAG). [2] We can compare the introduced definitions with Figure 2. The DAG starts with an empty exchange plan and sequentially adds cycles or chains. The external actions are represented using the blue arrows. Because of space constraints we only include one external action, at the end of the trajectory, in Figure 2. Once a maximal exchange plan is encountered, it transitions to a new KEP graph, modelled as a Markovian process $\mathcal{M}$ (see **Simulator** in section 6 for more details). The red box encloses a matching round (see section 5 for more details).

To suit the use of machine learning models, the states involved in our experiments are, in reality, computed from vertex embeddings of exchange plans. That is, given vertex embeddings $\{x_i\}_{i=1}^{|V|}$ in a vector space $\mathbb{R}^d$, we compute the averages $\frac{1}{|V(G)|} \sum_{i=1}^{|V(G)|} x_i$ and $\frac{1}{|V(H)|} \sum_{i=1}^{|V(H)|} x_i$ to obtain the embeddings of $G$ and $H$, respectively[3]. After the concatenation of these two vectors, we obtain the embedding of the state $s = (G, H)$ (i.e. the state corresponding to the exchange plan $H$). This map from an exchange plan (i.e. a graph) to its corresponding state will be denoted by $\sigma$.

**Definition 4.5** (Trajectory). A (complete) trajectory is a sequence $\tau = (s_0, s_1, \ldots, s_n)$ of states (here, $n \in \mathbb{N}$ is an arbitrary length) that satisfies $s_i \in child(s_{i-1})$ for $i = 1, \ldots, n$. Whenever $s_n \in \mathcal{X}$, we will refer to this (complete) trajectory as a **maximal trajectory**, or **maximal matching**.

In simple words, a trajectory corresponds to a sequential selection of exchanges for various KEP graphs (see Figure 2). The set of complete trajectories is denoted $\mathcal{T}$ and the set of trajectories following from the state $s$ is denoted $\mathcal{T}_s$.

---

[2]To prevent confusion, we reserve the use of the term "graph" solely for KEP graphs.
[3]The notation $V(G)$ and $V(H)$ is interpreted as the vertex sets of graphs $G$ and $H$.

**Parameterization.** Based on the terminology introduced above, we describe how to model probability distributions over states, partial and complete trajectories.

**Definition 4.6** (Forward and backward flow). The forward flow, denoted $P_F$, is a function $(s, s'), \theta \mapsto \mathbb{R}_+$ such that equation 1 is satisfied, where $\theta \in \mathbb{R}^d$ is a parameter vector. Similarly, the backward flow is a function $(s, s'), \theta \mapsto \mathbb{R}_+$ such that equation 2 is satisfied. By slightly abusing notation and dropping the dependence on $\theta$, we can use equation 3 to extend forward and backward flows to complete trajectories $\tau$.

$$\sum_{s' \in child(s)} P_F(s'|s, \theta) = 1 \tag{1}$$

$$\sum_{s \in par(s')} P_B(s|s', \theta) = 1 \tag{2}$$

$$P_F(\tau \mid s_0) := Z \prod_{i=1}^{n} P_F(s_i \mid s_{i-1}) \qquad P_B(\tau|s_n) := R(s_n) \prod_{i=1}^{n} P_B(s_{i-1} \mid s_i) \tag{3}$$

**Definition 4.7** (Initial flow). The initial flow $Z$ is a function $s, \theta \mapsto \mathbb{R}_+$, where $s$ is an internal state.

The parameters $\theta$ that are involved in $P_F$, $P_B$ and $Z$ are often omitted for simplicity. For example, in equation (6), we abuse notation and we drop the dependence of $P_F$, $P_B$ and $Z$ on the parameters unless necessary to disambiguate between two different $\theta$'s. In order to unify the notation between RL and GFlowNets, we make the identification in equation 4. The reward $R$ is a function of the external states $x \in \mathcal{X}$ and was described in detail in Definition 4.4 (see caption of Figure 2). We present the definition of the $Q$ function, as well as the detailed balance equation of the forward and backward probability transitions for GFlowNets in equation 6 of Definition 4.9. In equation 4.9, we omit the dependency of $R$ on the action since the end of an episode is an external state and thus, its action is trivial.

$$p_\theta(s'|s) := p_\theta(a \mid s) \equiv P_F(s' \mid s), \qquad a = (s, s') =: s \to s' \tag{4}$$

**Definition 4.8** ($Q$ function). The $Q$ function is a function $s, (s, s') \mapsto \mathbb{R}_+$. We make use of the identification found in equation 5, where we abuse notation a drop the dependence on the parameters.

$$Q(s \to s') := Q(s, a = s \to s') \tag{5}$$

**Definition 4.9** (Trajectory balance). The trajectory balance equation for $P_F$ and $P_B$ given a complete trajectory $\tau$ is as follows:

$$\frac{P_F(\tau \mid s_0)}{P_B(\tau|s_n)} = \frac{Z \prod_{i=1}^{n} P_F(s_i \mid s_{i-1})}{R(s_n) \prod_{i=1}^{n} P_B(s_{i-1} \mid s_i)} \tag{6}$$

If 0 over all trajectories, the trajectory balance loss indicates that the networks $P_F$ and $P_B$ correspond to a flow (see Bengio et al. (2023)). By sampling a set of trajectories, we optimize an estimator for the true loss. With the introduction of various definitions and identifications between RL and GFlowNet notation in mind, we will see in the next section how to learn policies over sequences of matching rounds (i.e. episodes). $Q$-learning, policy gradient, PPO, and GFlowNets each lead to their own loss that will be optimized over batches of episodes using an exploratory policy.

## 5  Generating episodes

In order to generate data over which the learning-based methods can fine-tune their parameters, we introduce a Markovian process $\mathcal{M}$ that generates KEP pools.

**Definition 5.1.** Let $L \in \mathbb{N}$ be arbitrary. An episode of $L$ matching rounds consists of a sequence $(\tau_i)_{i=1}^{L}$ of trajectories $\tau_i$ starting in an internal state $s_{0_i} = (G_i, \emptyset)$ and terminating in an external state $s_{n_i} \in \mathcal{X}$. We make use of the notations $s_0$ and $s_f$ for the artificial states corresponding to the start and final states of an episode. These are special purpose states that are neither internal nor external states, and that model the beginning and end of an episode. Furthermore, when the trajectories $\tau_i$ for $i = 1, \ldots, L$ are maximal, we call the episode a **static-matching episode**.

Each trajectory forming an episode corresponds to a matching round. The transition from trajectory $\tau_i$ to $\tau_{i+1}$ (i.e. from $s_{n_i}$ to $s_{0_{i+1}}$) indicates the end of a matching round and the start of a new one. In our scenario, we restrict episodes to static-matching episodes and the number of (maximal) trajectories contained in each episode to a fixed number $L$. Each episode has finite length and corresponds to a sequence of maximal matchings. In Figure 2, we can observe part of a static-matching episode. The matching round is contained inside the red box and we can see the previous and following internal states $(G', H')$ and $(G'', \emptyset)$, respectively. The choice of static-matching episodes is motivated by our goal to obtain near-optimal policies for traditional KEPs over multiple matching rounds. Therefore, the exchange plan selected at round $i$ needs to affect the matching round at time $i+1$, while the decision needs to be restricted to the current matching round. True dynamic KEP policies, where we obtain near-optimal solutions for a non-myopic setting, are reserved for future work and various approaches would be conceivable; nevertheless, learning-based approaches seem very well suited for this setting (see section 9). Using the notions introduced in section 4, the policies learned over episodes are dependent on parameter $\theta$. In the particular case of GFlowNets, the literature refers to this kind of learned GFlowNets as *conditional* GFlowNets (Bengio et al., 2023).

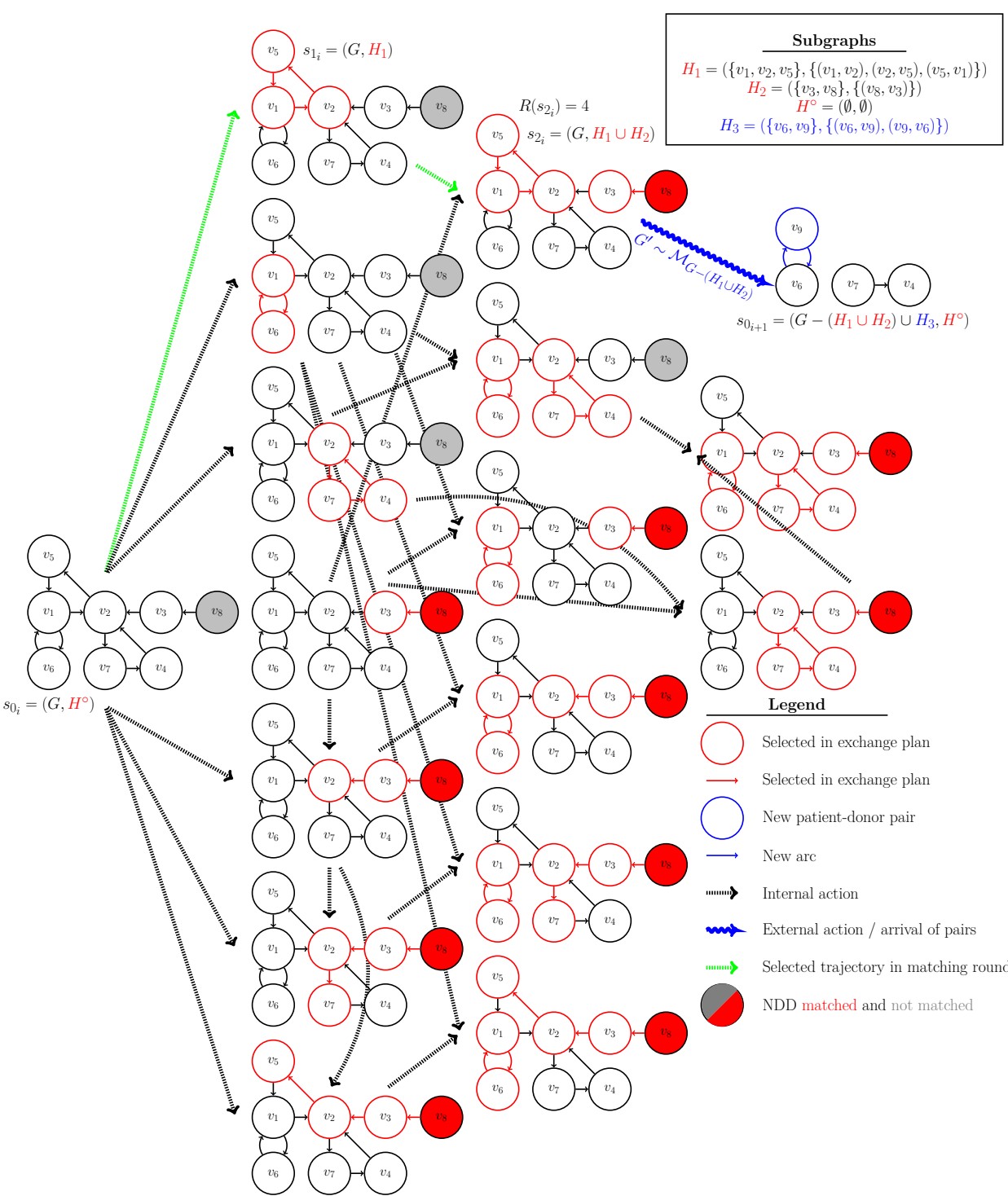

Figure 2: Example of a matching round. We start from the initial state represented with a KEP graph with an empty matching. At every step, we add cycles and chains to the exchange plan (red nodes/arcs) until we reach an external state (i.e. maximal matching). This external state transitions to a new initial KEP graph for the next matching round by following the markovian process $\mathcal{M}$ (blue curved arrow - only one is depicted due to space constraints). The green path corresponds to the trajectory for this matching round, with multiple trajectories forming episodes.

**Definition 5.2** (Exploratory policy)**.** An exploratory policy is a policy $\pi$ with full support over a domain $\{(\tau_i)_{i=1}^L : (\tau_i)_i \sim \mathcal{M}, \tau_i \in \mathcal{T}_{s_{0_i}}\}$. An exploratory policy over static-matching episodes is called a **static-matching exploratory policy**.

In the domain of the exploratory policy $\pi$, $\tau_i$ are trajectories forming episodes $(\tau_i)_{i=1}^L$ of length $L$. We can then evaluate the loss for our learning-based methods. Specifically, the GFlowNet loss $L$ for an episode is given by equation (7). Similarly, we can also obtain an estimate of the gradient for policy gradient in equation (8) and for the PPO loss (clipped with $\epsilon = 0.2$) in equation (9) [4].

The parameter $\theta$ corresponds to all the parameters of a neural architecture. For equations (7) and (9), we seek to minimize the loss $L$. In the case of GFlowNets, we obtain a flow loss since, if the loss is 0 over all episodes (or trajectories), we have a flow-matching condition called the trajectory balance equation (Malkin et al., 2022). This implies that $Z, P_F$ and $P_B$ form a flow. In practice, this loss will not be 0 and we will also not sum over all trajectories since there would be too many to compute. We will have to resort to sampling a subset of trajectories using $\pi$.

$$L(\theta) = \sum_{(\tau_k)_k \sim \pi} \left( \log \left( \frac{Z(s_0) \prod_{i_k=1}^{n_k} P_F(s_{i_k} \mid s_{i_k-1})}{R(s_n) \prod_{i_k=1}^{n_k} P_B(s_{i_k-1} \mid s_{i_k})} \right) \right)^2 \tag{7}$$

$$g = \mathbb{E}_{(\tau_k)_k \sim \pi} \left[ \sum_{i_k=1}^{n_k} \nabla_\theta \log P_F(s_{i_k} \mid s_{i_k-1}) Q(s_{i_k-1} \to s_{i_k}) \right] \tag{8}$$

$$L(\theta) = \mathbb{E}_{(\tau_k)_k \sim \pi} \left[ \sum_{i_k=1}^{n_k} \min \left( \frac{P_F(s_{i_k} \mid s_{i_k-1}, \theta)}{P_F(s_{i_k} \mid s_{i_k-1}, \theta_{\text{old}})} Q(s_{i_k-1} \to s_{i_k}), \right. \right.$$
$$\left. \left. \text{clip} \left( \frac{P_F(s_{i_k} \mid s_{i_k-1}, \theta)}{P_F(s_{i_k} \mid s_{i_k-1}, \theta_{\text{old}})}, 1 - \epsilon, 1 + \epsilon \right) Q(s_{i_k-1} \to s_{i_k}) \right) \right] \tag{9}$$

## 6 Experimental setup

We aim to validate our approach with empirical evaluations by addressing five core research questions formulated to demonstrate the capabilities of learning-based methods to iteratively build a variety of exchange plans for KEPs. To provide comprehensive answers to these research questions, we first outline the experimental setup and we describe the KEP instances simulator.

**Methods.** We first define the different suites of experiments that are performed. We evaluate generative flow networks (**GFlowNet**), *Q*-learning, policy gradient (**PolicyGradient**) and proximal policy gradient (**ProximalPolicyGradient**). We specify that the $Q$ function minimizes the squared difference between both sides of the Bellman equation as in the temporal difference learning (TD learning) loss. It is common in the GFlowNets literature to compare with methods such as **ProximalPolicyGradient** and **PolicyGradient** and we defer to this rich body of literature for examples. We evaluate learning-based methods on datasets of episodes sampled using our simulator: we simulate graphs using a Markovian process described below. The resulting static-matching episodes will allow us to model the evolution of the KEP pool over time as exchange plans are selected at various rounds, by more closely fitting the true distribution of possible graphs encountered over the full horizon ($L$ matching rounds).

In order to determine if the learning-based methods allow us to sample solutions that are close to optimal, we first need to compare the quality of our returned solutions against other methods that are not learning-based. We devise two heuristics to construct exchange plans. We will refer to the process of selecting sequentially disjoint exchanges uniformly at random to form a solution as **RandUniform**. The selection of disjoint

---

[4]We abuse notation and denote each method's loss and parameters as $L$ and $\theta$, respectively.

exchanges using a greedy process (i.e. largest first) will be referred to as **RandGreedy** (ties are broken uniformly at random). We also use an exact MIP solver (Gurobi 10.0.2) to compute an optimal solution in a single step to which we refer as **OptMIP**. Our mixed-integer implementation follows the hybrid position-indexed edge formulation found in Dickerson et al. (2016). We note that **OptMIP** will always return one solution in our experiments, in contrast to other sampling-based methods. Thus, the averages reported in Tables are rather single values for **OptMIP**. The baselines will serve to highlight the quality of the returned solutions and situate the effectiveness of the proposed learning approach.

**Simulator.** We simulate KEP graphs using a Markovian process. The simulator that is used in our experiments is base on the work of Saidman et al. (2006). Over multiple rounds of arrival, the number of incompatible pairs entering the pool is drawn from a Poisson distribution. For each of these pairs, the blood type of the patient and donor are drawn according to pre-defined probabilities. Compatibility arcs are then added between the newly introduced pairs and those already in the pool, as well as between the newly introduced pairs themselves, following the ABO compatibility model (Dean, 2005). For each of the new pairs, we randomly determine if the patient is *hard-to-match* given by their calculated panel reactive antibody percentage (cPRA; high means hard-to-match) (Tinckam et al., 2015). Then, outgoing arcs from each new pair are removed by drawing from a Bernoulli distribution with a parameter specified by the cPRA value. If the arrival rate at each round is $\lambda$ and we have $N$ rounds, the expected number of vertices in the KEP pool at the end is $\lambda N$. In our experiments, we did not simulate the arrival of non-directed donors to make the exposition simpler. Each graph (unless specified otherwise) in the distribution output by the simulator is obtained through $N = 20$ rounds with an arrival rate of $\lambda = 5$ (i.e., the expected number of vertices is 100).

**Generating episodes.** For each obtained graph, we simulate 1000 episodes by picking actions at random and follow the Markovian process between matching round ($N = 1, \lambda = 5$). Thus, we obtain in total a dataset $T$ consisting $100,000$ vectors $(\tau_1, \ldots, \tau_L)$ of trajectories $\tau_i$ forming episodes. We use conditional information (i.e. the initial graph at each matching round) as part of the input to our models by computing graph embeddings using the starting graph at each matching round.

**Model architecture.** The architecture that is used consists of three main pipelines: learning embeddings, learning the initial flow $Z$ (for GFlowNet) and learning the flow probabilities $P_F$. The layers that are used for each pipeline are given in Table 6 in Appendix A.

# 7 Computational results: evaluating the solution efficiency and quality of learning-based methods

Using our simulator, we can evaluate the performance of our learning approaches on static-matching episodes. This capability will allow us to model the evolution of the KEP pool over time as we follow the learned policy. Additional details regarding the architecture and other research questions can be found in Appendices A and C.

## 7.1 Quality of generated solutions

**RQ1: To what extent does the utilitarian value of solutions sampled from the learning methods align with those of the optimal solutions?** To answer this question, we sample 1000 solutions for each instance using the tuned networks, as well as using the baselines. For example, when evaluating **GFlowNet** on a single instance, we sample 1000 solutions given the single initial KEP graph, while for multiple instances, we sample 1000 solutions for each KEP graph. Given a sampling mechanism, we compare the samples that we generated and we select the best one (i.e. maximum utilitarian reward). The resulting samples should converge with probability 1 to their respective optimal value since each sampling mechanism has full support, provided the policy has full support. This is specifically the case for $\epsilon$-greedy policies.

We report the results from our experiment in Table 1, where the approximation ratio is equal to the utilitarian value of a method's returned solution divided by the optimal value. While the learning-based methods do not recover solutions that are as good as **OptMIP**, we do see a significant improvement over the heuristics. We note that among the learning methods, **GFlowNet** performs best. Further training of the networks,

| Method | Instance size ($\lambda N = 100$) | | | | | |
| --- | --- | --- | --- | --- | --- | --- |
| | **Single instance** | | | **Multiple instances** | | |
| | Best | Median | Worst | Best | Median | Worst |
| **OptMIP** | $1.000 \pm 0.000$ | $1.000 \pm 0.000$ | $1.000 \pm 0.000$ | $1.000 \pm 0.000$ | $1.000 \pm 0.000$ | $1.000 \pm 0.000$ |
| **RandUniform** | $0.873 \pm 0.040$ | $0.731 \pm 0.014$ | $0.492 \pm 0.035$ | $0.795 \pm 0.042$ | $0.681 \pm 0.018$ | $0.455 \pm 0.046$ |
| **RandGreedy** | $0.877 \pm 0.035$ | $0.779 \pm 0.030$ | $0.589 \pm 0.021$ | $0.821 \pm 0.037$ | $0.740 \pm 0.072$ | $0.492 \pm 0.011$ |
| **PolicyGradient** | $0.919 \pm 0.058$ | $0.826 \pm 0.016$ | $0.775 \pm 0.026$ | $0.838 \pm 0.067$ | $0.748 \pm 0.054$ | $0.684 \pm 0.292$ |
| **ProximalPolicy** | $0.923 \pm 0.065$ | $0.852 \pm 0.056$ | $0.787 \pm 0.033$ | $0.876 \pm 0.066$ | $0.759 \pm 0.022$ | $0.697 \pm 0.065$ |
| **$Q$-learning** | $0.904 \pm 0.059$ | $0.815 \pm 0.044$ | $0.768 \pm 0.034$ | $0.825 \pm 0.064$ | $0.725 \pm 0.061$ | $0.672 \pm 0.014$ |
| **GFlowNet** | $0.928 \pm 0.062$ | $0.854 \pm 0.064$ | $0.821 \pm 0.019$ | $0.897 \pm 0.051$ | $0.785 \pm 0.039$ | $0.702 \pm 0.0457$ |

Table 1: Average approximation ratio of learning-based methods against baselines. For each instance, we report the best, median and worst values. We average these values over all instances.

along with adjustments to the architecture and hyper-parameters, could potentially bring us even closer to the optimal value, offering promising opportunities for enhancing our method. Specifically, we envisage that training on KEP instances with $\lambda N > 50$ would help the performance when testing on larger KEP instances.

## 7.2 Scaling to larger KEPs

**RQ2: Is the generalization capability of RL and flow learning applicable to KEP instances of diverse dimensions?** In this experiment, we test the capability of RL and GFlowNets to generalize to larger graphs. Concretely, we train the neural networks on a distribution of smaller graphs and then, we evaluate the quality of the returned solutions on graphs of larger sizes.

| Method | Instance size ($\lambda N$) | | |
| --- | --- | --- | --- |
| | 50 | 100 | 200 |
| **OptMIP** | $1.000 \pm 0.000$ | $1.000 \pm 0.000$ | $1.000 \pm 0.000$ |
| **RandUniform** | $0.829 \pm 0.045$ | $0.795 \pm 0.045$ | $0.631 \pm 0.064$ |
| **RandGreedy** | $0.841 \pm 0.044$ | $0.821 \pm 0.034$ | $0.706 \pm 0.066$ |
| **PolicyGradient** | $0.914 \pm 0.055$ | $0.854 \pm 0.066$ | $0.727 \pm 0.072$ |
| **ProximalPolicy** | $0.933 \pm 0.056$ | $0.862 \pm 0.062$ | $0.742 \pm 0.066$ |
| **$Q$-learning** | $0.885 \pm 0.065$ | $0.829 \pm 0.056$ | $0.716 \pm 0.072$ |
| **GFlowNet** | $0.936 \pm 0.058$ | $0.872 \pm 0.057$ | $0.782 \pm 0.077$ |

Table 2: Average approximation ratio when training on instances of size (i.e. $\lambda N$) 50 and testing against instances of size 50, 100, and 200.

The goal of this experiment is to show the ability of a model to extend beyond the size of the graphs on which it was trained. Since large graphs are currently hard to train because of the large number of possible actions involved at each step, a possibility is to train on smaller graph instances and deploy the network on larger KEP graphs to obtain samples. We measure the quality of such samples on a network trained on instances with $\lambda N = 50$ (arrival of $\lambda = 5$ pairs each round in expectation; $N = 10$ rounds). In Table 2, we observe that when testing instances where the size of the initial KEP graphs is doubled, the learning-based methods still perform better than the heuristics. Inference was fast even on larger graphs; we were able to sample 1000 exchange plans per graph within a time of 15 minutes. The training time used was the same as in our prior experiments. While we do not report these values in Table 2, the approximation ratio of the best solution returned for larger graph sizes are surprisingly very similar to the ratios reported in Table 1 (Multiple instances).

**RQ3: How efficient is the generation of exchange plans?** In this experiment, we evaluate the ability of the learning approaches to sample multiple exchange plans efficiently. We compute the time to sample a

single solution averaged over 1000 solutions for each method, except for **OptMIP**, where we measure the time to return a single solution. The results can be found in Table 3.

| Method | Instance size ($\lambda N$) 200 |
|---|---|
| **OptMIP** | $16.812 \pm 0.374$ |
| **RandUniform** | $1.681 \pm 0.056$ |
| **RandGreedy** | $0.889 \pm 0.034$ |
| **PolicyGradient** | $1.702 \pm 0.075$ |
| **ProximalPolicy** | $1.527 \pm 0.090$ |
| $Q$-**learning** | $1.662 \pm 0.076$ |
| **GFlowNet** | $1.627 \pm 0.016$ |

Table 3: Aveage time (in seconds) to sample 1000 exchange plans. OptMIP time is for one solution, while learning-based and heuristic methods take the average over 1000 sampled solutions.

We observe that the slowest method to compute a single solution is **OptMIP**. The other heuristics or learning-based methods offer a more efficient method to obtain a large number of solutions. It has to be said that the learning-based approaches require the training of neural networks. The time spent to retrieve a solution is traded for the time spent learning the model. As the number and size of KEPs solved each year is high and expected to grow, the learning approach becomes viable, offering capabilities such as added flexibility in the choice of solution since we can generate many. It is also worth to mention that for larger KEP instances, **OptMIP** can often fail to return the optimal solution.

## 8 Computational results: evaluating the role of distributions in fairness and long-term policy behaviour

In this section, we will see that learned policies result in an efficient matching-generation mechanism that can be leveraged to generate distributions over exchange plans at each round. This ability combined with ideas from the fairness literature in KEPs will allow us to demonstrate the potential to mitigate the disproportionate probabilities of receiving a transplant over the set of patients in the pool. It will also allow us to better understand long-term impacts of KEP policies by directly modelling the evolution of the pool through the arrival and departure of patient-donor pairs.

### 8.1 Fairness in KEPs

**RQ4: Does the efficient generation of exchange plans allow us to mitigate fairness issues in KEPs?** In this experiment, we explore the extent to which learning approaches may improve fairness guarantees to patients by taking advantage of the fast exchange plan-sampling mechanism. Decision-makers can select solutions from a large set of exchange plans, from which they can devise lottery policies that satisfy various fairness criteria. The access to multiple high-reward policies ensure that they can balance utilitarian and fairness aspects as desired (see section 9 for a more in-depth discussion on this).

We choose to focus on the concept of individual fairness for KEPs (Farnadi et al., 2021). We begin by computing the set $P'$ of patients that can receive a transplant in the KEP. Using the subset $P'$, we make use of a fairness measure referred to as the $L_1$ distance for individual fairness (Farnadi et al., 2021).

**Definition 8.1** (Feasible matchings). For a graph $G = (V, A)$, we define the set of feasible matchings $\mathcal{X}_G$ as

$$\mathcal{X}_G = \{H \mid (G, H) \in \mathcal{X}\} \tag{10}$$

**Definition 8.2** ($L_p$ distance). Let $G = (V, A)$ be a graph and $P' \subseteq V$ the subset of vertices that *can* be in at least one solution from $\mathcal{X}_G$. The $L_p$-fairness of a probability distribution $\delta$ over $\mathcal{X}_G$ is given by

$$\sum_{v \in P'} (\delta_v - \bar{\delta})^p, \tag{11}$$

where $\delta_v := \sum_{H \in \mathcal{X}_G | v \in H} \delta(H)$, and $\bar{\delta} = \frac{1}{|P'|} \sum_{v \in P'} \delta_v$. The variable $\delta_v$ is the probability that vertex $v$ is in a selected exchange plan and $\bar{\delta}$ is the mean probability of being selected as part of an exchange plan for patients in $P'$. A probability distribution $\delta$ is *individually fair* if it minimizes the Objective 11[5].

This $L_1$ distance (here, $p = 1$) is a measure of the dispersion of patients' chances of being selected: a lower value indicates a more equally distributed chance of receiving a transplant, which is considered better. For each method, we sample 1000 maximal exchange plans and we compute the probability mass of the sampled exchange plans for two distributions: the uniform distribution and the distribution minimizing the $L_1$ distance. Because of the similarity in performance demonstrated amongst the learning-based methods in the previous experiments, we only include results for **GFlowNet** as they offered the best performance. We categorize our results as **GFlowNet w/ uniform** and **GFlowNet w/ IF**. We compare against **OptMIP**, whose $L_1$ score is computed from the distribution with a single solution. We also include the method of St-Arnaud et al. (2023), referred to as **Enum w/IF**, which consists of enumerating optimal solutions to KEPs using a column generation procedure in order to minimize the $L_1$ value. For the column generation procedure, we set a time limit of 1 hour. We also added the baseline heuristic **RandGreedy** to evaluate the capacity of learning-based methods to enhance fairness over a simple solution sampling process. In order to understand how individual fairness would scale, we also used different values of $N$ (with the same $\lambda$) to generate 1000 instances for each additional $\lambda N$ combination. We report our results in Table 4, where we group results by the value of $\lambda N$ (i.e. the expected size of the instances in the generation process).

| | Instance size ($\lambda N$) | | |
| Method | 50 | 100 | 200 |
|---|---|---|---|
| **OptMIP** | $15.381 \pm 1.814$ | $43.265 \pm 8.318$ | $81.211 \pm 14.484$ |
| **Enum w/ IF** | $10.532 \pm 1.489$ | $25.423 \pm 3.991$ | $61.306 \pm 8.153$ |
| **GFlowNet w/ uniform** | $11.794 \pm 1.130$ | $34.467 \pm 4.651$ | $63.597 \pm 9.426$ |
| **GFlowNet w/ IF** | $10.701 \pm 1.280$ | $26.981 \pm 3.823$ | $62.049 \pm 8.005$ |
| **RandGreedy** | $14.440 \pm 1.791$ | $39.814 \pm 5.295$ | $75.869 \pm 12.655$ |

Table 4: Average individual fairness measures per instance size.

We observe that GFlowNets (and learning-based methods) improve measures of fairness when compared with **OptMIP** as they allow for a more varied subset of patients to be included in its distribution of solutions. Furthermore, the learning-based methods outperform the heuristic **RandGreedy**, indicating that these methods are helpful in improving individual fairness measures. The optimal values achieved by **Enum w/ IF** are very close to the ones offered by **GFlowNet** minimizing the $L_1$ with respect to the sampled exchange plans (referred to as **GFlowNet w/ IF** in Table 4). In fact, making use of learning-based methods reveals to be a promising approach as they result in efficient samplers *once trained* that improve individual fairness for larger instances. We can even manage to generate a large number of solutions for instances of size $\lambda N = 500$ and 1000, while it is not feasible for **OptMIP** or **Enum w/ IF**. In fact, for such large KEP graphs, we could not even manage to generate a single solution in the time limit. We note that individual fairness has been shown to suffer from a technical issue when optimized in conjunction with a utilitarian (St-Arnaud et al., 2023) because empty exchange plans are optimal in terms of the fairness metric. This is, however, not the case in our experiments since we sample solutions that are maximal, i.e., they cannot be extended further. This experiment serves to highlight how the ability to generate a large set of exchange plans can be used to mitigate fairness issues through lottery policies. In practice, decision-makers might wish to balance fairness and utilitarian approaches. The capacity to explore the space of lottery policies through the efficient generation of exchange plans provides fertile ground for the exploration of these ideas.

By generating multiple solutions through reinforcement and flow learning methods, we can sample among them and use the resulting patient probabilities to empower decision-makers with vital information related to the structure of the KEP graph that would otherwise be missing. For example, in Figure 3, we can observe how these probabilities are distributed. Note that with **GFlowNet w/ IF**, the probabilities are less unevenly distributed. It is then possible to identify patients that are "mathematically" hard-to-match

---

[5]For more context, see St-Arnaud et al. (2023).

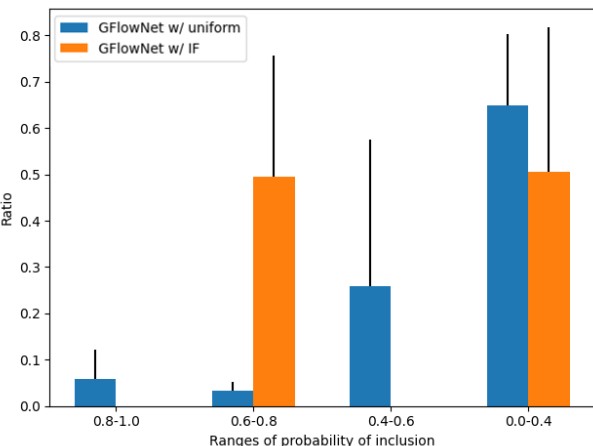

Figure 3: Distribution of patient transplant probabilities by ranges of probabilities. Values are averaged over all instances with $\lambda N = 100$.

according to the KEP graph, as a complement to clinical definitions involving their cPRA measure (Dickerson et al., 2014; Saidman et al., 2006).

Another conceivable method to jointly optimize the number of transplants and some fairness objective would be to make use of the conditioning nature of approach, where we condition on the weights over both objectives (see Jain et al. (2023) for an application to GFlowNets). These weights can later be changed to evaluate the effect on the space of solutions returned by the model, offering some added flexibility to decision-makers. In this work, we focus on the use of a single objective because of the added level of complexity these conditional models entail. In the case of individual fairness and patient probabilities, multi-objective RL approaches and multi-objective GFlowNets cannot provide any help as they are functions of distributions of exchange plans.

## 8.2 Long-term policy behaviour

**RQ5: Can efficient solution generation be used to evaluate the long-term impact of a selection policy?** In this experiment, we deploy a trained GFlowNet (on instances of size $\lambda N = 50$) and we perform multiple matching rounds using it. Concretely, in each round, an exchange plan is selected, the selected vertices are removed from the graph and the arrival of new pairs with their corresponding compatibility arcs is simulated. We use the same arrival rate $\lambda$ and number of arrival rounds $N$ for the incoming data. In total, we generate 1000 static-matching episodes corresponding to multiple matchings and changes in the pool. In Table 5, we compare the use of **GFlowNet** in each round with the use of **OptMIP**. The computation times reported are with respect to the solution sampling time. Times to build the MIP (for **OptMIP**), to enumerate available actions and to obtain the tensors (for **GFlowNet**) are not reported as they are similar for both methods.

Foremost, we are interested in predicting the evolution of the KEP pool over multiple matching rounds, as using **OptMIP** is slow and, for large real-world sizes, impractical. For this purpose, we deploy a GFlowNet on a horizon of length 10 (i.e. 10 matching rounds). At each matching round, we select the maximum-sized exchange plan from the set sampled by the GFlowNet (1000 exchange plans for each matching round). Since the GFlowNet approximates the true optimal distribution with respect to the utilitarian objective, the probability that the selected exchange plan is optimal goes to 1 as the number of samples increases at each round.

We compare the cumulative number of transplants divided by the number of matching rounds against **OptMIP**. The averages were computed over 1000 trajectories, much as in the previous experiment. We refer the reader to Figure 4 for results. The efficient sampling mechanism offered by GFlowNets allow us to

| Instance size $\lambda N$ | 50 | | |
|---|---|---|---|
| Method/Matching rounds | 2 | 5 | 10 |
| **OptMIP** | $43.862 \pm 5.057$ | $118.639 \pm 13.517$ | $189.771 \pm 29.428$ |
| processing time (s) | $15.219 \pm 0.393$ | $38.910 \pm 1.990$ | $81.629 \pm 6.240$ |
| **GFlowNet** | $40.580 \pm 7.291$ | $101.431 \pm 12.674$ | $144.615 \pm 29.630$ |
| processing time (s) | $1.328 \pm 0.012$ | $3.263 \pm 0.031$ | $5.103 \pm 0.062$ |
| Instance size $\lambda N$ | 100 | | |
| Method/Matching rounds | 2 | 5 | 10 |
| **OptMIP** | $71.006 \pm 9.874$ | $168.305 \pm 19.171$ | $292.203 \pm 36.743$ |
| processing time (s) | $35.604 \pm 1.17$ | $93.791 \pm 7.29$ | $199.603 \pm 17.709$ |
| **GFlowNet** | $66.672 \pm 11.116$ | $149.503 \pm 22.395$ | $229.646 \pm 39.645$ |
| processing time (s) | $3.336 \pm 0.018$ | $8.965 \pm 0.046$ | $18.385 \pm 0.091$ |
| Instance size $\lambda N$ | 200 | | |
| Method/Matching rounds | 2 | 5 | 10 |
| **OptMIP** | — | — | — |
| processing time (s) | — | — | — |
| **GFlowNet** | $136.013 \pm 28.223$ | $296.246 \pm 28.018$ | $633.861 \pm 75.446$ |
| processing time (s) | $3.910 \pm 0.045$ | $10.704 \pm 0.147$ | $23.731 \pm 0.284$ |

Table 5: Average number of transplants for multiple matching rounds when training on instances of various sizes 50 and testing against instances of the same size. The average processing times reported are with respect to sampling 1000 solutions. No instances were solved for **OptMIP** within the time limit for $\lambda N = 200$.

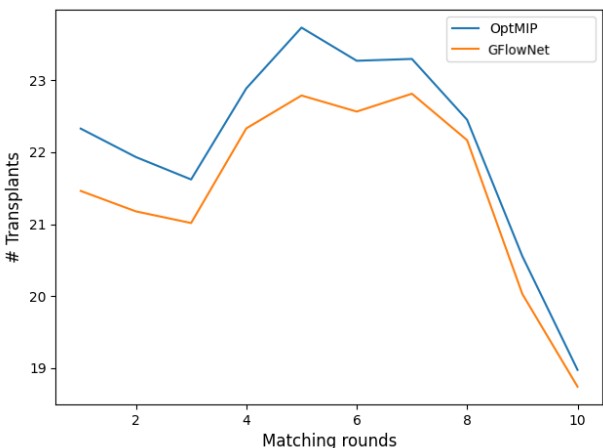

Figure 4: Average cumulative number of transplants after each round, divided by the number of rounds over instances of size $\lambda N = 50$.

approximately compute the expected number of transplants performed over multiple rounds of KEPs under a MIP framework. This allows us to model the evolution of the KEP pool under uncertainty over large instances, and Figure 4 demonstrates that we can approximate the expected number of transplants achieved under **GFlowNet** to high precision. The repeated optimization for MIP proves too costly to evaluate over instances of size 200, as shown in Table 5 (dashed lines), which serves to underscore the effectiveness of our sampler in predicting the evolution of the KEP pool.

## 9 Discussion, future directions, and broader impacts

In this work, we have demonstrated the ability of learning-based methods, to complement exact MIP approaches in KEPs. Our learned models, as efficient samplers of near-optimal solutions, give us two highly valuable outcomes. Firstly, we are able to efficiently provide multiple high-quality solutions that can be used, for example, to promote fairness or to offer alternative solutions in the event of unexpected infeasibility (e.g. donor drop-off). Secondly, we can approximate the long-term impact of choosing a particular objective (matching policy) on the KEP pool. In the future, we can learn non-static-matching (i.e. dynamic) policies over a longer horizon of simulated arrivals to approximate the optimal policy over multiple matching rounds, given an objective (e.g. utilitarian objective). Validating our approach with hierarchies of objectives and other real-world complexities in KEPs would be of interest. Of particular interest, a thorough exploration of the space of lottery policies can offer decision-makers the ability to explore policies that provide certain fairness guarantees while remaining efficient in terms of the expected number of transplants. The learning-based mechanism we present in this article can be used to efficiently generate multiple exchange plans that can be included in the support of these lottery policies, scaling to large kidney exchange pools. We envisage the extension of our conditional learning-based method to a family of objectives as in Jain et al. (2023), in order to explore a large set fair policies as well as policies that balance utility and fairness. Our explorations need not be confined solely to KEPs. The applicability of learning-based methods to other matching or combinatorial problems remains a promising avenue for future study. Particularly in the context of policy-making, they have the potential to provide guidance on the combinatorial space of solutions and to help anticipate long-term impacts.

With respect to the broader impacts that our work can have, we highlight the expanded applicability of learning-based methods to address combinatorial problems, with a specific emphasis on KEPs. Our findings showcase promising results in learning (near-)optimal distributions that can guide policymakers and clinicians, and offer patients supplementary information regarding solution selection and kidney allocation based on their graph positions. This work represents an initial step towards acquiring deeper insights into a novel approach for solution selection based on distributions of solutions and the approximation of long-term effects in problems that wield significant impact on individuals' lives. While our initial results are encouraging, the ethical, legal, and social consequences of our approach need in-depth consideration to ensure responsible implementation. In addition, our approach is currently tailored to KEPs. Extending it to broader combinatorial problems requires careful validation and adaptation. The real-world scenarios given their dynamic and multifaceted nature, specifically in health care, involve complexities that our approach may not fully capture. For instance, how patient preferences or health conditions evolve over time or how health-care policies change. Finally, our evaluations rely on simulated data and as the approach evolves, obtaining diverse and comprehensive real-world datasets becomes crucial. However, we acknowledge the difficulty of accessing such datasets due to their sensitive nature.

## 10 Acknowledgements

This work was funded by the NSERC grant 2024-04051 and 2021-04378, Canada CIFAR AI Chair, and Google scholar award. This research was enabled in part by support provided by Calcul Québec (www.calculquebec.ca) and Compute Canada (www.computecanada.ca).

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
