## A  Model architecture

Our model architecture for the GFlowNet is summarized in Table 6. All other learning-based methods use a similar architecture to the $P_F$ pipeline for their policy and $Q$ function (or derived advantage function as in PPO and policy gradient). The first main component of the model consists in obtaining the vertex embeddings of a graph. We aggregate the resulting vectors by summing over the vertices and call the resulting vector, the *graph embedding*. For a subgraph $H$ of $G$, we sum over the vertices included in the subgraph. As input to the flow pipeline, we concatenate the graph embedding $G$ with the embedding of the subgraph $H$ corresponding to the current matching (i.e. selection of exchanges). To compute the output flows (resp., probabilities) $P_F$ for a particular trajectory, we need to normalize the flows (resp., probabilities). This normalization step is not done for the $Q$-function as values can be arbitrary. Given a subgraph $H$ of $G$, we consider all available exchanges $c$ that can be added to $H$ and perform a LogSoftmax operation over them. We sum the resulting probabilities over a trajectory to obtain the joint probability (logarithmic scale).

| Embedding Pipeline | |
|:---:|:---:|
| Count | Layer type |
| 1 | {Vertex: Embedding(4, 32), Edge: Embedding(1, 32)} |
| 2 | NNConv(32, 32, Identity) $\rightarrow$ reLU $\rightarrow$ GRU(32, 32, 2) |
| **Initial flow ($Z$) Pipeline** | |
| Count | Layer type |
| 1 | Linear(32, 32) $\rightarrow$ reLU |
| 2 | Linear(32, 1) |
| **Forward Flow ($P_F$) Pipeline** | |
| Count | Layer type |
| 1 | Linear(32, 32) $\rightarrow$ reLU |
| 2 | Linear(32, 32) $\rightarrow$ reLU |
| 3 | Linear(32, 32) $\rightarrow$ reLU |
| 4 | Linear(32, 1) $\rightarrow$ reLU |

Table 6: Architecture of the flow networks for KEPs

## B  Wasserstein distance

In this section, we define the mathematical program that allows us to estimate the Wasserstein distance $W_2$ using the distribution $\mu_2$ of optimal exchange plans (solutions optimal for **OptMIP**) and the distribution $\mu_1$ of sampled exchange plans using one of the following methods: **GFlowNet**, **RandUniform** and **RandGreedy**.

Suppose that we have enumerated optimal solutions. We create a linear program using the state embeddings of each solution of $\mu_1$ and $\mu_2$. This linear program (LP) is described in equations (12)-(15). The variables $\mu_1(x)$ and $\mu_2(y)$ are created for every state embedding $x$ and $y$ part of the support of $\mu_1$ and $\mu_2$, respectively. Note that $\mu_1$ and $\mu_2$ are merely the probability distributions given through enumeration of optimal solutions for **OptMIP** and the sampling mechanisms respectively; they do not reflect the true distributions and are merely proxies for them. The coefficients $\|x - y\|_2$ are computed before generating this LP from the state embeddings of the enumerated (i.e. the $y$ vectors) and sampled exchange plans (i.e. the $x$ vectors). Here, $x$

and $y$ are also used as index sets to distributions $\mu_1$, $\mu_2$ and $\mu$.

$$\min_{\mu} \sum_{(x,y)} \|x - y\|_2 \mu(x,y) \tag{12}$$

$$\text{s.t.} \sum_{x} \mu(x,y) = \mu_2(y) \qquad \forall y \tag{13}$$

$$\sum_{y} \mu(x,y) = \mu_1(x) \qquad \forall x \tag{14}$$

$$\sum_{x,y} \mu(x,y) = 1 \tag{15}$$

Since above we have a linear program, we can easily use column generation for the enumeration of solutions. In other words, we can solve the linear program (12)-(15) with a subset of solutions and iteratively add columns (solutions) as needed, i.e., as long as there are variables with negative reduced cost.

## C  Additional experiments

### RQ6: How evident is the quality of learned (near-) optimal distributions through neural networks for KEPs?

We test the ability of the training phase to recover the shape of the distribution of optimal distributions with the use of the Wasserstein distance between the two distributions.

In this experiment, we measure the distance between the distribution of optimal solutions to KEPs and the distribution output by a GFlowNet. We use the Wasserstein metric $W_p$ (Kantorovich, 1960). This metric is also known as the "earth mover's distance". It computes a joint probability distribution $\mu$ whose marginals are the two probability distributions $\mu_1, \mu_2$ measured by the metric. The joint probability that it computes minimizes the integral $\int \|x - y\|_p \mu(x,y)$ for $p \geq 1$. This forms the optimal transport plan from one distribution to the other. If the two distributions are equal (i.e. $\mu_1 = \mu_2 = q$), we have $W_p(q,q) = 0$.

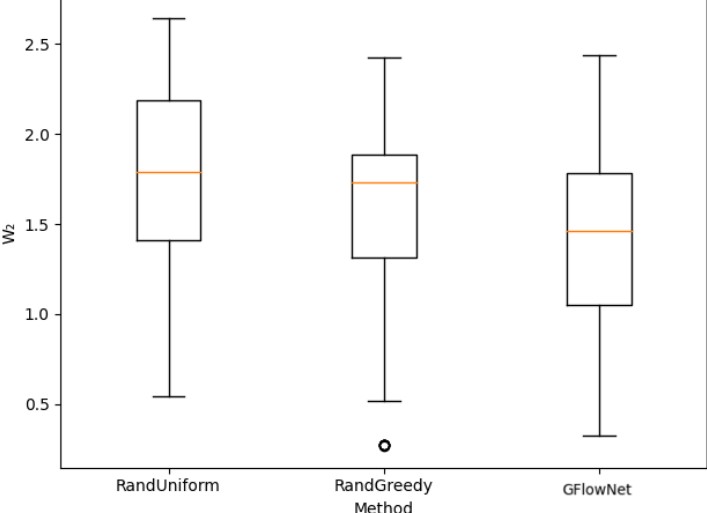

Figure 5: $W_2$ distance between the distribution of optimal solutions given by **OptMIP** (target) and each method. The circles represent outliers

Since the number of optimal solutions and the number of trajectories are large for instances with $\lambda N = 100$, we must resort to an approximation of the true Wasserstein metric. We enumerate optimal solutions with

column generation (we stop when no improvement is observed) and we evaluate the probability that this trajectory is outputted by the GFlowNet. We can also sample many trajectories using our GFlowNet and find their corresponding probabilities. Under the target distribution, these trajectories should get 0 probability if they are not optimal. Using all the sampled and enumerated solutions, we can form a mathematical program that we optimize to compute the Wasserstein distance (see appendix B).

We can see in Figure 5 that the method which returns the smallest $W_2$ distance on average is **GFlowNet**. It is worth noting that given the current reward function, we do not expect to get a $W_2$ that is equal to 0, even if the GFlowNet perfectly learns the distribution proportional to $R$. One could, for example, use a reward $R(G) = e^{\beta |P(G)|}$ with $\beta > 0$ being larger than 1 (see also Figure 2). This would imply that the peaks of the resulting distribution more closely resemble the optimal distribution of the target (optimal) distribution found through **OptMIP**.