# OpenReview forum: "A Learning-Based Framework for Fair and Scalable Solution Generation in Kidney Exchange Problems"
_TMLR — Accepted by TMLR_

### Review · Reviewer_4Wqt · 2025-01-16

**Summary Of Contributions:**

Kidney exchange problems (KEPs), as a combinatorial problem, typically require computationally expensive methods such as mixed-integer program (MIP) solvers to solve for exact optimal solutions. As an alternative, this paper proposes a learning-based approach for KEPs: training a stochastic policy to generate exchange plans using algorithms such as reinforcement learning and generative flow networks (GFlowNets) over simulated KEPs, then sampling multiple exchange plans using the learned policy on new KEPs. The motivation of this approach is to find approximate solutions to KEPs using a method that can generate multiple exchange plans efficiently.

The paper explores a range of research questions through experiments on simulated KEPs. The experimental results show that the learning-based approach can provide exchange plans that achieve about 90% of the optimal utilitarian reward by selecting the best of the multiple exchange plans generated by the learned policy. This performance is higher than that achieved by heuristic-based baselines, RandUniform and RandGreedy. In addition, it is shown that the learning-based methods, after training, can generate exchange plans more efficiently than the exact solution solver based on MIP, which makes then more viable for simulating longer-term KEPs. The paper also discusses potential applications of the learning-based approach for fairness considerations in KEPs, based on its capability to generate multiple exchange plans.

**Audience:**

Yes

**Broader Impact Concerns:**

The broader impacts of the work have been discussed in the last section of the paper.

**Claims And Evidence:**

No

**Requested Changes:**

* Expand Section 7.1 to include additional measures of solution quality beyond the utilitarian reward achieved by the single best exchange plan generated. For example:
  - The distribution of rewards over all generated plans.
  - The mean/median utilitarian reward.
  - The variation in the best-sample reward across different sample sizes. As the authors note in Section 7.1, all random methods, including the heuristic-based methods, can theoretically achieve near-optimal rewards if they have full support on all valid exchange plans and the sample size is sufficiently large. It would be interesting to evaluate how the best-sample reward of different methods changes with varying sample sizes.

* In Section 8.1, the discussion of fairness improvements should be coupled with an evaluation of the utilitarian rewards achieved by different methods. This would provide a more complete picture for understanding the trade-offs between fairness and the utilitarian rewards (the number of transplants) achieved by each method.

* Question: The caption of Table 3 states that it shows the average time to sample 1000 exchange plans. However, it is also mentioned that only one solution is generated for OptMIP. Does the time reported for OptMIP correspond to generating a single solution, while the times for other methods represent the total time required to generate 1000 solutions?

**Strengths And Weaknesses:**

## Strengths

* While applying learning-based methods to find approximate solutions for combinatorial problems is an existing line of research, this paper contributes by proposing an RL- and GFlowNets-based approach for KEPs—a combinatorial problem with real-world impact.

* The paper is generally well organized. The problem setting is formulated clearly, and the visualizations provided with the concrete examples are helpful for understanding the sampling procedure of exchange plans in the proposed approach.

* When selecting the highest-utility exchange plan out of 1000 plans generated, the proposed learning-based methods are shown to generate exchange plans that achieve about 90% of the optimal utilitarian reward achieved by the exact solver, outperforming heuristic-based baselines RandUniform and RandGreedy.

* The learning-based methods, while requiring a training process before use, are able to generate exchange plans more efficiently with their learned policy than the exact MIP solver. This could be valuable for simulating multi-round KEPs and thus studying the evolution of KEP pools over a longer time horizon.

* The demonstrated efficiency, coupled with the 90%-optimal best-sample reward, highlights the potential of the learning-based approach as a practical and faster approximate solver for KEPs.

## Weaknesses

* **Limitation of the comparison of solution quality:** In Section 7.1, the learning-based methods and the heuristic-based methods generate multiple (1000) exchange plans for a single KEP graph. However, only the best exchange plan with the highest utilitarian reward is compared across methods, while there is no comparison of the distribution of utilitarian rewards over all generated exchange plans. Furthermore, summary statistics such as average/median utilitarian rewards for each method are not included. One of the advantages of the learning-based approach, as argued in the paper, is its capability to compute a diverse set of (near-)optimal solutions. However, this is not well supported by the current results, as only the best solution generated is evaluated. Additionally, the utilitarian rewards of other generated exchange plans matter since all generated exchange plans are used as the support for the KEP policy in the discussion related to fairness improvements in Section 8.1.

* **Limitation of the discussion on fairness improvements (Major):** Without evaluating the utilitarian rewards (number of transplants) achieved by different methods, the evaluation and discussion of individual fairness (IF) measures alone in Section 8.1 do not provide sufficient insights into the performances of the compared methods. For the baseline *Enum w/ IF*, the distribution of solutions has a support that only contains optimal solutions, therefore achieving the optimal utilitarian reward. In contrast, there is no evaluation of the utilitarian rewards achieved by the proposed learning-based methods *GFlowNet w/ uniform* and *GFlowNet w/ IF*. Comparing methods based solely on the IF metric is insufficient since they may lead to very different utilitarian rewards. A hypothetical policy (e.g., a uniformly random policy) could achieve a low IF measure while not being practically useful for KEPs if it results in very few transplants. Therefore, the claim that the learning-based approach improves fairness in KEPs requires additional evaluation of the corresponding utilitarian rewards.

---

> ### Author Response · Authors · 2025-02-19
> **Response to weaknesses**
>
> - The goal of this research question was to show that learning-based methods can recover at least one solution that is close to optimal. In Table 1, we added the worst solution and the median of the sampled solutions, averaged over all instances. We amended the caption of the table to reflect this change. We believe that this shows to the reviewer that on average, the number of transplants is still higher than when using the heuristics. While it is true that the utilitarian rewards of all solutions in the distribution matter, depending on the fairness scheme used, it might be necessary to consider suboptimal solutions with a larger optimality gap. Indeed, it is possible that certain patient-donor pairs are never found in higher utility solutions and therefore a decision-maker might wish to select solutions that include these pairs with a certain probability even if this is suboptimal in terms of the total number of transplants. Arguing in favour a specific frequency and tolerance for sub-optimality is left at the discretion of decision-makers and we make no claims as to what should ethically be implemented.
>
> - The **Enum w/ IF** method described in our experimental section does not contain only optimal solutions but rather uses the method of St-Arnaud et al. [2023] to generate distribution of solutions that optimize both the $L_1$ metric and the total number of transplants simultaneously through a bi-objective conic program. For the evaluation of the **GFlowNet w/ uniform** and **GFlowNet w/ IF**, we instead refer the reviewer to the updated version of Table 1, where we list the worst solution sampled and the median. It is possible to see that the price of fairness for learning-based methods is never prohibitively high.

---

> ### Author Response · Authors · 2025-02-19
> **Response to requested changes**
>
> - ``Expand Section 7.1 to include additional measures...`` In the updated version of Table 1, we report the best, median and worst approximation ratios for the distribution of sampled solutions. The variation of best-sample reward for varying sample sizes surprisingly does not vary much from the ratios reported in Table 1. We added a mention of this fact in the last paragraph of RQ2 in the last sentence.
>
> - ``In Section 8.1, the discussion of fairness improvements should...`` With the addition of the median and worst approximation ratios in Table 1, it is possible to observe the extent to which the number of transplants vary in sampled solutions. Section 8.1 only showed that it is possible to improve individual fairness by taking advantage of the varied pool of solutions. In practice, decision-makers can seek to balance utility with fairness as in St-Arnaud et al. [2023]. From the values of Table 1, we observe that the price of fairness is not prohibitively large for learning-based methods.
>
> - ``Question: The caption of Table 3 states that it shows the average...`` This is true that OptMIP only has one solution, hence the "aver-
> age" is rather a single value. For other methods, the average reported is over the 1000 solutions generated. To remove this ambiguity, we added a clarification in the caption of the table. We also added a clarification in the paragraph above the "Simulator" section to stress that OptMIP values are always with respect to a single solution.

---

### Review · Reviewer_g4z3 · 2025-01-20

**Summary Of Contributions:**

The authors propose learning-based methods for the kidney exchange problem, where the goal is to find reallocation cycles between incompatible patient-donor pairs. Their methods will in each episode focus on the static setting, in which the set of patient-donor pairs is fixed and given. Between different episodes, they model an arrival process in which new patient-donor pairs are added to the pool.

The authors perform a computational study to compare the performance of the learning-based methods with traditional solution methods for the kidney exchange problem, such as a MIP formulation and simple heuristics (e.g., greedy). While the solutions generated by the learning-based methods have no guarantee of optimality (where optimality is measured as the maximum number of transplants), the authors report on the quality of the generated solutions.

**Audience:**

Yes

**Broader Impact Concerns:**

/

**Claims And Evidence:**

Yes

**Requested Changes:**

See the points in the section on weaknesses of the paper under the previous question.

Additional small comments:

-	P.3: kidney exchange is not only implemented in countries in Europe, as stated, but also widely used in the US and Canada, for example
-	P.3: To the best of my knowledge, it is common to say that an arc (u,v) means that the patient from pair v is compatible with the donor of pair u, and not the other way around as written in the paper. See, e.g., the paper ‘Novel integer programming models for the stable kidney exchange problem’ by Klimentova et al. (2022).
-	P.3, bottom section 2: maximum predetermineD exchange length.
-	P.4: “The drawback of their methodology is that fairness objectives are an input, effectively restricting the landscape of solutions made available to the policy-makers.” I don’t understand this comment. These papers aim to optimize individual fairness criteria (e.g., minimum selection probability) when allowing for all optimal exchange plans. In what sense do they restrict the solutions that are made available?
-	P.5 (middle): “[if the exchange plan is maximal, we can select it] … or we can augment it with additional cycles and chains.” How can we add additional cycles or chains when the exchange plan is already maximal?
-	P.5, Def. 4.3: sloppy math: T is a mapping so add : between T and (s,a). Later on in the paper, the notation for mappings is also different from convention. And what should be before =(s,s’)?
-	P.6: “Actions are represented by blue arrows”. There is only one blue arrow?
-	P.8: in Fig. 2, pair v_2 is included both in H_1 and in H_2? They can only be included in one cycle/chain, no?
-	P.9 (top): learning-based methods instead of leaning-based (also p. 13)
-	P.10: $\tau_i$ instead of tau_i
-	P. 14: “Since the GFlowNet approximates the true optimal distribution, …” Optimal distribution with respect to what objective function? Will this be a distribution optimizing an individual fairness measure? Or do you refer to the number of matches in the matchings it finds?
-	RQ3: why should you compare to generating 1000 optimal solutions using OPTMIP, and not just to comparing one optimal solution? Because the 1000 sampled solutions using the proposed methods will at the first step be used to find the one with the maximum number of transplants, which is exactly what you obtain by solving the MIP formulation to optimality once. (Side question: will 1000 sampled solutions using your methods all be different? Because MIP will generate 1000 different solutions, which is more challenging than simply sampling 1000 times).

Lastly, the references have some missing information and lack consistency.:
-	Gao (2019): add additional information (arxiv?)
-	Some references have doi (Abraham et al., 2007), others do not
-	Demeulemeester et al. (2024) no page numbers or issue number
-	Bengio (2021a) is published

**Strengths And Weaknesses:**

STRENGHTS:
- To the best of my knowledge, applying learning-based methods to the kidney exchange problem is new. This is a relevant problem that is widely studied in the Operations Research and Economics literature, and traditional solution methods have been highly focused on Mixed Integer Programming (MIP) formulations.
- The authors introduce the terminology that is required for the learning-based methods in an accessible way that is understandable for readers who have limited experience with learning-based methods.

WEAKNESSES:

-	First, I believe that this is a setting where optimality of a solution is extremely important. For all evaluated instances with 200 pairs, for example, Table 2 illustrates that all heuristic methods (incl. learning-based) have an approximation ratio of less than +- 78%. For illustration, imagine the optimum would be 150 matches, for example. In that case, this would mean that adopting one of the discussed heuristics instead of an exact solution would result in 33 less transplantations! These are 33 patients that could have received a life-saving transplantation, but do not get one because of a choice not to focus on trying to find the best possible solution. The paper argues that MIP methods are not scalable. While it is true that computation times increase rapidly for larger instances, there is a large literature that proposes alternative MIP formulations for various kidney exchange variants. The paper: ‘On the kidney exchange problem: cardinality constrained cycle and chain problems on directed graphs: a survey of integer programming approaches’ by Mak-Hau (2017), for example, illustrates that instances with up to 2000 pairs could be solved optimally, and since then, newer methods have been developed.

-	The discussion on individual fairness is lacking in my opinion. First, only the criterion of L_1 fairness is considered, which is not a good measure to compare methods that find distributions over solutions of different quality. Imagine a solution in which no exchanges are executed. The average probability is zero, and the L_1-fairness is zero as well, obtaining a better score than all methods presented in the paper. It could be that the learning-based obtain a L_1-fairness score close to the optimal because they allow for worse solutions, while the optimal distribution only uses solutions that find an optimal number of exchanges. As the learning-based methods optimize over a different (larger) solution space, they could have well obtained a lower L_1-fairness score. To truly claim that the learning-based methods improve the individual fairness of the problem, they should be compared based on other individual fairness metrics (e.g., leximin, which maximizes the minimal selection probability, Nash product, L_2 norm…?). Additionally, a comparison with, for example, RandGreedy could be added to convince the reader that the learning-based methods outperform a very simple heuristic.

-	Second, the improvement in individual fairness with respect to OPTMIP is not a merit of the learning-based methods, but is simply due to the fact that OPTMIP only returns a single solution. Papers such as Farnadi et al., 2021, St-Arnaud et al., 2023, and Demeulemeester et al., 2025 already observed that a distribution over multiple optimal solutions greatly outperforms selecting one optimal solution deterministically with respect to individual fairness.

-	It is not stated to which MIP formulation the learning-based methods are compared to. The choice of the MIP formulation matters a lot for the time required to find a solution.

-	The claim that finding distributions over optimal exchange plans using MILPs is not scalable is unfounded, in my opinion. On page 4 (top), it is claimed that this is only scalable for instances up to 50 instances. In Demeulemeester et al. (2025), for example, methods are introduced to find various distributions that do not rely on full enumeration of the optimal exchange plans. The solution times that are presented by them are around 1 second for some of the distributions for instances up to 70 pairs.

---

> ### Author Response · Authors · 2025-02-19
> **Response to weaknesses**
>
> - We agree with the reviewer about the extreme importance of
> using exact methods allowing to determine optimal exchange plans. We
> seek to stress that the methodology proposed here does not aim to replace
> MIP methods but to complement them:
>   - in the abstract: "(...)provides a complementary landscape to their traditional integer programming-based toolbox for promoting fairness and societal benefit."
>   - in the conclusions (section 9): "In this work, we have demonstrated
> the ability of learning-based methods, to complement exact MIP approaches in KEPs." Indeed, in section 1, we identify the goals of our
> methodology to be related with having an efficient sampling mechanism of high-quality solutions. This allow us, in section 8, to showcase the value of our methodology, notably
>     - to mitigate fairness issues. Some patients might never be in such a globally optimal solution. Decision-makers might deem it necessary to consider sub-optimal exchange plans to clear such patients from the pool occasionally.
>     - to evaluate the long-term impact of a selection policy. This is possible due to our efficient sampler which has the ability to quickly sample multiple solutions for large KEPs.
>
>   In section 7 of the revised paper, in the last paragraph of RQ1, we added a sentence commenting on how to potentially improve performance in terms of the utilitarian objective for our learning-based methods.
>
> - In our experiments, the solutions returned by the learning-based methods are guaranteed to be maximal exchange plans (see page 5, Section 4, second sentence of the paragraph "Preliminaries"). Hence, the issue described with respect to the L1 metric, i.e. an empty exchange plan, does not apply. We have highlighted in the text of RQ4 that we are indeed only sampling maximal exchange plans (see paragraph after definition 8.2). We also included a comparison with RandGreedy to further underscore the improvement over a basic heuristic. We show that this heuristic, although it leads to fairness improvements in comparison with OptMIP, results in smaller fairness improvements than learning-based methods (second sentence after Table 4).
>
> - In our experiments, the Enum w/ IF approach follows the methodology of St-Arnaud et al. [2023] to enumerate solutions and construct a distribution over them using a column generation process, optimizing for a fairness objective. The merit of the learning-based approach that we aim to stress is on its ability to be close to the optimal result from St-Arnaud
> et al. [2023], while scaling better to large instances. When deploying Enum w/ IF on larger instances with λN ∈ {500, 1000}, we could not find a single solution in the column generation procedure. We added this fact in the paragraph after Table 4, before the paragraph discussing Figure 3.
>
> - This is the broadly used hybrid position-indexed edge formulation (HPIEF) detailed in Dickerson et al. [2016]. We have added a clarification in the paper mentioning and citing the relevant work in the Experimental setup (paragraph before the simulator, after introduction of OptMIP).
>
> - The sentence mentioned by the reviewer is about the method of Farnadi et al. [2021] that does not scale beyond 50 pairs. For other
> methods that do not require full enumeration, e.g. the individual fairness in St-Arnaud et al. [2023], it is possible to scale to larger instances, e.g. 200 pairs as we see in table 4 under the label Enum w/ IF. However, such approaches do not scale much beyond that: in the time limit that we imposed in the experimental section, no KEP instance of more 500 vertices could be solved, while it is possible to efficiently sample solutions for learning-based methods. About the paper by Demeulemeester et al. (2025), they use a column generation approach as in St-Arnaud et al. [2023], so similar scalability limits are expected.

---

> ### Author Response · Authors · 2025-02-19
> **Response to additional comments**
>
> - ``kidney exchange is not...`` We provided Europe as an example. To better reflect the worldwide adoption of these programs, we have included mentions to other KEP programs and cited them accordingly in the first paragraph of Section 2.
>
> - ``To the best of my knowledge...`` This was a typo and indeed, this is the convention that is used in this paper. We corrected this and cross-referenced the rest of the article to ensure that there were no further incorrect use of the order.
>
> - ``bottom of section 2...`` Thank you for indicating this typo. We have corrected it in the text.
>
> - ``“The drawback of their methodology is that...`` Before solving the optimization related to the kidney exchange problem, a choice has to be made to determine the fairness scheme (objective function) that is used to determine fair solutions. We do not claim that our work is an improvement over this property as we simply use a reward modelled on the total number of transplants in a solution. However, one potential strength of using learning-based approaches is to condition on, say, different weights for each patient-donor pair in the objective. This would enable decision-makers to explore the behaviour of selecting different fairness schemes by simply fine-tuning the weights on which the model is conditioned. This idea is discussed in the paragraph before section 8.2 of our article and it is indeed very important explore the efficiency and usefulness of such as process. Such a methodology is behind the exploration of the Pareto frontier in works such as Jain et al. [2023]. We rewrote the sentence referred by the reviewer in section 3 to make it more clear.
>
> - ``“[if the exchange plan is maximal, we can select it] …`` We assume that the reviewer is referencing the sentences "the current exchange plan is maximal or not. In the latter case, we can opt
> to select this exchange plan under our policy, or we can augment it." We wish to clarify that when building a solution, we extend it with additional cycles and chains as long as it is non-maximal. We reformulated this sentence to make it clearer as the construction is nonstandard.
>
> - ``Def. 4.3: sloppy math...`` We thank the reviewer for pointing out this mistake. The math notation should instead be $(s, a)  \mapsto s'$. Secondly, there was a missing
> term before the $=$ sign. The equation should have been $a = (s, s')$, i.e. the action a is a tuple equal to $(s, s')$, the transition from $s$ to $s'$. These mistakes have been corrected in the revised paper.
>
> - ``“Actions are represented by blue arrows”...`` The are external actions for all external states but due to space limit and in order to show which exchange plan is selected, we only represent one external action in blue. We have added a clarification to underscore this fact in the Preliminaries of Section 4 and in the caption of the figure.
>
> - ``in Fig. 2, pair v_2...`` This is indeed a typo. As the image suggests, the graph $H_2$ should not contain the vertex $v_2$ as there is not arc $(v_3, v_2)$ that is drawn in red. We thank the reviewer for this observation. We also removed the unnecessary parentheses for the vertex sets of the graphs.
>
> - ``learning-based methods`` We fixed this typo in the main text.
>
> - ``... instead of tau_i`` We fixed this error to $τ_i$.
>
> - ``“Since the GFlowNet approximates the true optimal...`` It is implied to be optimal with respect to the total number of transplants. Since we also discuss various other objectives in the article, we agree that this should be made explicit and we added a mention to the corresponding objective in the article.
>
> - ``RQ3: why should you compare to generating 1000...`` OptMIP only returns one solution and not 1000 as suggested. We added clarifications in the paragraph before the "Simulator" section to make this idea less ambiguous. As suggested in the following point, it is indeed interesting to ask whether the sampled solutions are diverse or not. Indeed, we never obtained the same solution twice in our sampling procedure, while this is not technically impossible.
>
> - ``Lastly, the references have some missing information...`` We fixed these issues in the references and also proceeded to thoroughly reverify other entries in the bibliography for typos and newer versions.

---

### Review · Reviewer_r3iH · 2025-02-05

**Summary Of Contributions:**

The paper proposes to solve kidney exchange problems with GFlowNets by training it with RL or the trajectory balance loss.
The authors claim that their approach can complement traditional integer programming methods, enhance fairness, and provide a more efficient way to simulate the evolution of KEPs over time.

**Audience:**

Yes

**Claims And Evidence:**

No

**Requested Changes:**

- Figure 1 has notation or vocabulary issues. v is both used to denote a pair (as P is a set of incompatible pairs) and to denote single donor (v_8). It seems to me that P is the set of incompatible patients (not a set of pairs).

- Please confirm what are the backbone networks used in Policy Gradient, PPO & Q-Learning? Notably for the Q network that seems to be used in every method.

- The definition of the Q function (4.8) is not clear and not complete, are you maximizing a discounted sum of rewards?
If so, it looks like comparing algorithm that maximize \sum r_t (policy gradient, Q-Learning & PPO) vs ones that maximize the final reward r_T (GFlowNet loss (7)) is not right. Please describe how is the Q function learned or estimated?

- It looks like you are training in an offline RL scenario according to the Section 6. (Generating episodes). Therefore, why would you use online RL algorithms in this setup? Please clarify.

- "It is also worth to mention that for larger KEP instances, OptMIP can often fail to return the optimal solution." The claim is not supported by experiment results. Can you find large instances where the solution you obtain is better than OptMIP?

**Strengths And Weaknesses:**

Strengths:
- the usage of Gflownet seems appropriate in this scenario

Weaknesses:
- lack of clarity:
  * In Figure 1, if N is the set of *non-directed* donors, why v_8 has a link with v_3 only?
  * Providing an intuition of internal state and external state on top of the formal definition would help the reader
  * The definition of the transition function is not clear. For a state (G, H), it should be explained how the next state are constructed.
  * R(s_n) in (6) is not defined without an input action.
  * Intuition about the Trajectory balance loss would also help, what is the motivation for using this loss?
  * see requested changes

- The experiments on fairness is not very convincing as you are comparing a method that do not optimize for fairness (OptMIP), with a method that do (GFlowNet followed by a selection according to fairness, GFlowNet w/ IF).
If GFlowNet w/ uniform do not explicitly optimize fairness, it's often the case in multi-objective optimization that when you are a bit worst on one objective you can be better on another.
A cleaner experiment would present results displaying the Pareto front and include the performance of OptMIP if it would also optimize for fairness.

- when it comes to KEP, it seems that obtaining slow optimal solutions is preferred over obtaining fast sub-optimal ones

---

> ### Author Response · Authors · 2025-02-19
> **Response to weaknesses**
>
> - ``lack of clarity``
>   - Non-directed donors are donors that altruistically register in the program without an associated patient. An arc (link) from a non-directed donor to an incompatible pair is created if the non-directed donor is compatible with the patient of that incompatible pair. In the example of Figure 1, the non-directed donor represented by vertex $v_8$ is only compatible to the patient of the incompatible pair represented by vertex $v_3$. Hence, $v_8$ has a single out-going arc.
>
>   - In the article, we mention that an internal action "will correspond to cycles and chains that can be included to augment the exchange plan in the internal state", while an external action "corresponds to the selection of the exchange plan as a matching." External states are thus maximal matchings, while internal states are exchange plans in the process of being built. We have added a clarification after their definition in the text immediately before definition 4.2.
>
>   - The transition function simply corresponds to adding cycles and chains for internal states and transitioning to a new KEP graph after a matching round for external states. We have added this clarification immediately after the definition.
>
>   - We define $R$ to be a state-action function but for GFlowNets, at the end of the trajectory the action is trivial and the same for all terminal states (in our setting). For this reason, we omit the dependency on the states as an abuse of notation and to better conform to the working notation. We mention this is the revised paragraph above equation (6).
>
>   - If $0$ over all trajectories, the trajectory balance loss indicates that the networks PF and PB correspond to a flow (see Bengio et al. [2021]). By sampling a set of trajectories, we optimize an estimator for the true loss. We have added this clarification in the last paragraph of section 4 (after definition 4.9).
>
> - ``The experiments on fairness is not very convincing...`` OptMIP is included as a baseline where no fairness optimization is performed. Additionally, we included the **Enum w/ IF** by St-Arnaud et al. [2023] which uses mixed-integer programming to optimize a fairness objective. In this way, (i) in Table 4, by noting that our learning-based methods are close to the average individual fairness of **Enum w/ IF**, we highlight their value as a mechanism to promote fairness, (ii) in Table 1, by nothing that our methods can provide solutions with a high utilitarian value, we also show their value in terms of one of the most important objectives in practice, and (iii) in the text of RQ4, by adding the revised insights on experiments for larger instances, we conclude that the learning-based methods scale well, while the performance of the **Enum w/ IF** deteriorates due to large running times (see paragraph after Table 4 and before Figure 3, where we fail to obtain a single solution for instances with $\\lambda N \\in \\{500, 1000\\}$).
>
> - ``when it comes to KEP...`` In practice, the primary KEP objective is indeed to obtain a single solution, optimizing the utilitarian objective, which can be achieved through MIP methods. However, prior work on fairness has been shown to require the enumeration of multiple solutions in order to optimize fairness metrics [Farnadi et al., 2021, St-Arnaud et al., 2023, Demeulemeester et al., 2025]. In addition, these solutions may not be optimal with respect to the utilitarian objective, as there can be patients that are not part of any utilitarian-optimal solution. The space of feasible solutions is therefore extremely large. Exploring it can be computationally prohibitive, especially as exchange pools grow in size.
>   The focus of this paper is to explore whether learning-based methods can efficiently recover high-quality solutions after training, complementing combinatorial optimization methods. Our results demonstrate that our approach performs well in terms of the utilitarian objective (Table 1), fairness (Table 4), and integration on the evaluation of long-term impacts of selection policies (Table 5). Importantly, as exchange pools increase in size, MIP methods may become computationally unpractical for application of fairness and long-term analyzes, whereas our learning-based approach remains scalable and efficient. This makes our method particularly valuable for real-world decision-making, where decision makers can still use MIP approaches to obtain a single utilitarian-optimal solution and then compare with a fairness sampling and long-term analyzes coming from the learning-based approaches. They can then decide the best balance and compromise between the obtained solutions.

---

> ### Author Response · Authors · 2025-02-19
> **Response to requested changes**
>
> - ``Figure 1 has notation or vocabulary issues...`` The letter $v$ is used to denote vertices in the graph. The vertices can both represent patient-donor pairs (set $P$ ) and non-directed donors (set $N$ ). The vertex set $V$ of a KEP graph can be written as the disjoint union of $P$ and $N$ . We added this mathematical clarification in section 2, after the introduction of sets $P$ and $N$ from Figure 1.
>
> - ``Please confirm what are the backbone networks...`` All networks have the same base architecture as described in Table 6 to sample actions. The weights are of course updated according the different objectives. We specified in the paragraph of Appendix A that $Q$ functions use the same architecture as the forward probabilities without the normalization step.
>
> - ``The definition of the Q function (4.8) is not clear...`` The $Q$ function minimizes the difference between both sides of the Bellman equation as in the temporal difference learning (TD learning) loss. GFlowNets do not maximize the final reward, but rather minimize a loss in order to sample solutions proportionally to the reward. It is common in the GFlowNets literature to compare with methods such as PPO and we defer to this rich body of literature for examples. We amended the paragraph in ``Methods'' (section 6) to clarify the points raised in this question.
>
> - ``It looks like you are training in an offline RL...`` While we generated a fixed dataset prior to training, we wish to highlight that the parameter updates were done in an online manner over the full episode. In other words, the state-action values and probabilities were bootstrapped using updates from prior state-action pairs in the episode. We used this approach as it led to better stability during training and while we could have fully implemented an online learning approach with fresh data for each episode, training was faster with a fixed dataset. The only exception is, of course, in the case of GFlowNets, where we updated the parameters at the end of a complete episode.
>
> - ``"It is also worth to mention that for larger KEP instances...`` **OptMIP** is optimal with respect to the utilitarian objective. We
> only reported values for instances that **OptMIP** was able to fully optimize. However, if we increase the size of the KEP graphs, we eventually hit a threshold beyond which we cannot recover a single solution in the time
> limit. For example, KEP pools of size $\\lambda N \\in \\{500, 1000\\}$ did not return a single solution in the column generation.

---

### Author Response · Authors · 2025-02-19
**Message intended for all reviewers**

We would like to thank each individual reviewer for their thorough analysis of our work. We respond below to each individual point as clearly and directly as possible. We hope that our responses will address any concern that the reviewers raised. Where appropriate, we also included new results that we added to the paper. Any changes to the paper are referenced by page or section number and we also specify surrounding text and/or location to help the reviewers identify the relevant areas.

---

### Comment · Action_Editor_xqty · 2025-03-15
**A followup question for the authors**

As I am preparing my recommendation, I wanted to follow up with one technical question which I think gets at some reviewer concerns.  When are rewards given in your model?  Definition 4.4 specifies the reward on an arbitrary state transition, but comments after Definition 4.7 suggest it is intended to apply only to external states, i.e. the s_n at the end of a trajectory.  This also makes more intuitive sense to me because otherwise the first matching or cycle chosen shows up in the reward at every step.

---

> ### Author Response · Authors · 2025-03-17
> **Clarification for definition 4.4**
>
> It is indeed the case that the reward should only be applied to states at the end of a trajectory as they correspond to the selection of a particular exchange plan. For this reason, the definition should rather be $R(s,a) = \exp^{|P(H’)|}$, where $a = ((G, H), (G’, H’))$ and $H’$ is a maximal matching.  It is missing to incorporate the latter detail. Based on this comment, we modified definition 4.4 to make this clear.

---

### Public Comment · ~Arthur_Maffre1 · 2025-12-09
**n the validity of conditional GFlowNets without importance sampling in finite-horizon non-ergodic settings**

Dear authors,
Thank you for this interesting application of conditional GFlowNets to dynamic kidney exchange problems.
I recently uploaded a short working paper that analyzes some theoretical aspects of the conditional GFlowNet setup used in your work, particularly:

the violation of the full-support assumption required for unbiased condition-decomposable losses (Bengio et al., 2023, Eq. 77) in finite-horizon settings with non-ergodic graph dynamics,
the resulting bias toward low-round / low-complexity pools,
quadratic gradient scaling in the trajectory balance objective under restricted marginal support,
and non-stationary amortization with propagating gradient variance.

The note is available here: https://www.researchgate.net/publication/398493734_Bias_in_Conditional_GFlowNets_without_Importance_Sampling_Violation_of_Full-Support_Requirement_Quadratic_Gradient_Scaling_and_Non-Stationary_Amortization_with_Propagating_Gradient_Variance_in_KEP_set

I believe these points may explain some of the observed phenomena in Figure 4 (strong performance in early rounds, degradation in later rounds) and raise questions about the theoretical validity of the approach without off-policy corrections or importance weighting.
I would be very interested in your thoughts on these remarks, and whether importance sampling or infinite-horizon approximations were considered during development.

Best regards,
Arthur Maffre
Independent researcher (DIRO, Université de Montréal)

---

### Decision · Action_Editor_xqty · 2025-03-19

**Recommendation:** Accept with minor revision

**Comment:**

Some revision is needed to address the Claims and Evidence criterion.

One substantive way to address this, in the spirit of reviewer suggestions, would be to consider these various methods in the 2D space of efficiency vs fairness operating points and provide a method by which GFlowNets can be used to trace out a curve in this space rather than merely identifying two possible operating points.  For example, can the approach be used with a parameterized objective to choose a distribution to optimize a weighted combination of fairness and efficiency?  (This might also clarify the somewhat mysterious role played by the version with the uniform distribution which currently is not really discussed.  How should we think about it?  As an ablation?)

Alternatively, the TMLR guidelines identify another pathway to ensure the claims made in the paper are supported: “Another is simply for the authors to adjust (reduce) their claims.”  I believe it would be accurate to use language such as “suggests the potential of GFlowNets to be used to address fairness”, since I believe the results do show enough as is to demonstrate potential, although the efficiency of the new methods as is should still be reported in either Table 4 or some alternative form.  Suitable caveats and discussion of the research challenges for future work around implementing this would need to be added, for which comments and suggestions from reviewers should be incorporated.

Either approach would be adequate to satisfy the criteria, so I leave the choice to the authors.  The first would make for a stronger paper, but I recognize it requires some additional work.  I believe the second falls well within the expectations for a minor revision.

**Audience:**

The reviewers unanimously agree in their recommendations that this is met and I agree.  Readers interested in either KEP or GFlowNets may find things of interest to them.  I believe this holds true regardless of the resolution of the issue of the claims about fairness.

**Claims And Evidence:**

This criterion has more disagreement about whether the result in the paper are enough to make the claims “convincing.”  Many of the concerns here have been addressed, including confirming that rewards only occur after the full matching is sampled, making the methods used in this paper comparable to traditional RL approaches. However the reviewers have identified one key place where this standard is not currently met: the claim that “by using exchange plans generated through policy learning, we demonstrate how to improve individual fairness.”

The reviewers rightly point out that simply showing that this method produces fairer results than a fairness unaware baseline and achieves fairness comparable to a method that optimized fairness as a secondary objective doesn’t build a convincing case that these methods can reasonably be used to improve fairness.  Normally there is an expectation that the nature of the tradeoff implied by the algorithm be examined, even if the suitability of any given operating point is something best determined by stakeholders.  Currently, this material does not examine efficiency at all, leaving the efficiency effects of applying the IF weighting unknown beyond the crude estimates available from using the “worst” column of Table 1.

---

> ### Author Response · Authors · 2025-04-14
> **Minor revisions**
>
> Based on the two different options regarding the revisions to our paper, we decided that revising the section on fairness to adjust our claims is the preferable option. The other option requires exploring the Pareto front of a multi-objective space, where we train conditional GFlowNets on the weights to individual objectives. Although this should be done in future research, this task is tantamount to an entirely new paper. A complete experimental pipeline needs to be built to obtain results, while it would also require a significant amount of computing resources. The results of such an endeavour would also stand on their own and their focus would deviate from our current work significantly: our contributions reflect points 1 and 2, which are methodological in nature, as well as point 3 outlined in the abstract; the discussion on fairness falls under point 3, serving to illustrate the practical relevance of learning-based frameworks in the context of fairness and motivates future work on the combination of such objectives with the utilitarian one. For this reason, we agree that lowering and adjusting our claims to make this objective clearer is warranted. We proceeded to revise the section on fairness (section 8) to better reflect this position. In the list below, we detail specific modifications to the text that were introduced to address the minor concerns raised by the reviewers.
>
> * In the first paragraph of section 8, we amended the text to better reflect the potential improvement to fairness: “This ability combined with ideas from the fairness literature in KEPs will allow us to demonstrate the potential to mitigate the disproportionate probabilities of receiving a. transplant over the set of patients in the pool.”
>
> * In RQ4, we changed the text to underscore that the efficient generation of exchange plans can be used to select a fairer outcome for the patient pool: “we explore the extent to which learning approaches may improve fairness guarantees.” We also added a few sentences that clarify our position with respect to the improvement to fairness measures: “Decision-makers can select solutions from a large set of exchange plans, from which they can devise lottery policies that satisfy various fairness criteria. The access to multiple high-reward policies ensure that they can balance utilitarian and fairness aspects as desired (see section 9 for a more in-depth discussion on this).” In the previous sentences, we shift the emphasis from our achieved performance towards a blueprint that can be imitated to explore the space of dual utiliatarian-fair lottery policies.
>
> * At the end of section 8.1, we add a few sentences to highlight the fact that our experiment shows the potential of our method to search the space of lottery policies to find high-reward policies that also mitigate fairness issues: “This experiment serves to highlight how the ability to generate a large set of exchange plans can be used to mitigate fairness issues through lottery policies. In practice, decision-makers might wish to balance fairness and utilitarian approaches. The capacity to explore the space of lottery policies through the efficient generation of exchange plans provides fertile ground for the exploration of these ideas.”
>
> * In the conclusion, we add a few sentences to highlight the fact that the exploration of the space of lottery policies lies in the hands of decision-makers or future researchers. It is thus made clear that our experiments were provided as motivation and a proof-of-concept for this inquiry: “Of particular interest, a thorough exploration of the lottery policy space can offer decision-makers the ability to explore policies that provide certain fairness guarantees while remaining efficient in terms of the expected num-
> ber of transplants. The learning-based mechanism we present in this article can be used to efficiently generate multiple exchange plans that can be included in the support of these lottery policies, scaling to large kidney exchange pools. We envisage the extension of our conditional learning-based method to a family of objectives as in Jain et al. (2023), in order to explore a large set of fair policies as well as policies that balance utility and fairness.”